# Efficient intervention for pulmonary fibrosis via mitochondrial transfer promoted by mitochondrial biogenesis

Ting Huang [1], Ruyi Lin [1], Yuanqin Su [1,2], Hao Sun [1], Xixi Zheng [1,2], Jinsong Zhang [1], Xiaoyan Lu [1], Baiqin Zhao [3], Xinchi Jiang[1], Lingling Huang [4], Ni Li [5], Jing Shi [6], Xiaohui Fan [1,7], Donghang Xu [4], Tianyuan Zhang [1,2,8] ✉ & Jianqing Gao [1,2,4,8,9] ✉

The use of exogenous mitochondria to replenish damaged mitochondria has been proposed as a strategy for the treatment of pulmonary fibrosis. However, the success of this strategy is partially restricted by the difficulty of supplying sufficient mitochondria to diseased cells. Herein, we report the generation of high-powered mesenchymal stem cells with promoted mitochondrial biogenesis and facilitated mitochondrial transfer to injured lung cells by the sequential treatment of pioglitazone and iron oxide nanoparticles. This highly efficient mitochondrial transfer is shown to not only restore mitochondrial homeostasis but also reactivate inhibited mitophagy, consequently recovering impaired cellular functions. We perform studies in mouse to show that these high-powered mesenchymal stem cells successfully mitigate fibrotic progression in a progressive fibrosis model, which was further verified in a humanized multicellular lung spheroid model. The present findings provide a potential strategy to overcome the current limitations in mitochondrial replenishment therapy, thereby promoting therapeutic applications for fibrotic intervention.

Mitochondria are the organelles that control the homeostasis, bioenergy, metabolism, biosynthesis, and apoptosis of eukaryotic cells[1–3]. Recent studies have revealed additional functions of mitochondria as signaling organelles that participate in cell stress sensing, cell dynamics, protein secretion, and cell ageing[4–8]. Such critical roles of mitochondria in functioning cells make mitochondrial dysfunction becoming a considerable pathogenesis[9–11]. For example, several previous studies have indicated that the dysfunction of mitochondria in lung cells is one of the main causes of

promoting pulmonary fibrosis (PF)[12,13]. In addition, a recent study further has shown that mitochondrial quality has a significant impact on PF progression[14]. Therefore, the repair of dysfunctional mitochondria has been proposed as a strategy for disease intervention[15]. Among all the strategies available to correct mitochondrial functions, mitochondrial replenishment therapy (MRT), which uses exogenous mitochondria to replenish dysfunctional mitochondria, is one of the most promising. Accumulating evidence has supported the effectiveness of MRT in reconstructing the

[1]College of Pharmaceutical Sciences, Zhejiang University, 310058 Hangzhou, China. [2]Hangzhou Institute of Innovative Medicine, College of Pharmaceutical Sciences, Zhejiang University, 310058 Hangzhou, China. [3]Department of Thoracic Surgery, The Second Affiliated Hospital, Zhejiang University School of Medicine, 310009 Hangzhou, China. [4]Department of Pharmacy, The Second Affiliated Hospital, Zhejiang University School of Medicine, 310009 Hangzhou, China. [5]Department of Cardiothoracic Surgery, Ningbo Medical Centre Lihuili Hospital, Ningbo University, 315041 Ningbo, China. [6]School of Pharmaceutical Sciences, Hangzhou Medical College, 311300 Hangzhou, China. [7]National Key Laboratory of Chinese Medicine Modernization, Innovation Center of Yangtze River Delta, Zhejiang University, 314102 Jiaxing, China. [8]National Key Laboratory of Advanced Drug Delivery and Release Systems, Zhejiang University, 310058 Hangzhou, China. [9]Cancer Center, Zhejiang University, 310058 Hangzhou, China. ✉e-mail: tianyuanzhang@zju.edu.cn; gaojianqing@zju.edu.cn

damaged mitochondrial homeostasis and restoring cellular functions[16,17].

Highly efficient mitochondrial delivery with high specificity towards mitochondrial dysfunctional cells is the key to the success of MRT. To date, mitochondrial delivery approaches can be broadly categorized into direct and indirect delivery routes[15]. The former involves directly transplanting isolated mitochondria into the disease sites locally, which poses challenges such as the lack of selectivity in identifying mitochondrial dysfunctional cells, and the fast deactivation of the isolated mitochondria in the extracellular environment[15]. Conversely, an indirect delivery route exploits a more recently discovered ability to spontaneously donate mitochondria from normal cells to injured cells[17,18], showing the major advantages of avoiding the additional isolation of mitochondria and protecting the activity of transferred mitochondria. Thus, this naturally occurring phenomenon provides a new avenue for using living cells as both a donor and carrier of mitochondria for MRT.

However, a major challenge in the indirect therapeutic strategy is finding optimal mitochondrial carrier cells, which should have the ability to migrate to diseased tissues and recognize dysfunctional cells, and initiate intercellular mitochondrial delivery. Furthermore, it is preferable that these carrier cells have low bioenergetic needs and possess the powerful ability to generate mitochondria, which supports efficient and continuous mitochondrial exportation. Nevertheless, such a demand seems to counteract the nature of cells. Generally, cells with powerful mitochondrial generation capacities, such as myocardial cells, also have high bioenergetic requirements[19]. Conversely, mesenchymal stem cells (MSCs) have low bioenergetic needs in their glycolytic state[20,21], making them more willing to donate their mitochondria to bioenergy desiderated cells[17,22]. However, the low energy demands of MSC often restrict their mitochondrial generation, thereby limiting the number of mitochondria available for transportation and adversely affecting therapeutic outcomes. For example, we had previously observed an efficient intervention for PF using MRT with MSCs engineered by iron oxide nanoparticles (IONPs). However, these engineered MSCs demonstrated poor therapeutic efficiency against the progressive PF, partly due to the insufficient and quickly exhausted mitochondrial transfer[23]. In fact, the importance of an efficient and continued supply of mitochondria to restore the mitochondrial bioenergetics of injured lung epithelial cells (LECs) has been described in the interventions for PF[24].

The present study aimed to determine the possibility of promoting mitochondrial biogenesis in low-energy demanding MSCs, as well as the therapeutic benefits of these mitochondrial-increased MSCs in the treatment of progressive PF. We observed that careful treatment of human placental-derived MSC (hMSC) with pioglitazone (Pg), a prescription drug for diabetes, could efficiently activate the mitochondrial biogenesis in hMSC primarily through the PGC-1α–NRF1–TFAM pathway. Furthermore, joint engineering using Pg treatment and IONPs stimulation (Pg-Fe-hMSC) could achieve highly efficient and sustained intercellular mitochondrial delivery of the engineered hMSC, which we termed high-powered hMSC. Consequently, these high-powered hMSCs demonstrated a prominent therapeutic ability to intervene in PF progression in a mouse model with progressive PF (Fig. 1). Furthermore, the efficient therapeutic potential of the high-powered hMSCs was confirmed in both fibrotic human lung cells and three-dimensional (3D) humanized lung spheroids.

## Results

### Pioglitazone promoted mitochondrial biogenesis of hMSC and assisted intercellular mitochondrial transfer

The potential of Pg to induce mitochondrial biogenesis in hMSC was investigated first. The results demonstrated that mitochondrial biogenesis in hMSC was severely restricted by both the concentration and the duration of Pg treatment. Only after treating hMSC with Pg at a concentration of 10–15 μM for 5 days was a significant increase in mitochondrial biogenesis observed; whereas this improvement in mitochondrial biogenesis was modest under the other conditions (Supplementary Fig. S1). Because PGC-1α is considered a dominant regulator of mitochondrial biogenesis[25,26], the bioeffects of Pg on PGC-1α expression levels were studied next. As expected, Pg treatment significantly elevated the PGC-1α expression in both hMSC and Fe-hMSC (Fig. 2a). To fully understand the mechanism underlying the enhanced Pg-induced mitochondrial biogenesis, two downstream factors in the PGC-1α pathway, including nuclear respiratory factor 1 (NRF1) and mitochondrial transcription factor A (TFAM), which drive the production of mtDNA and increase mitochondrial mass[27], were evaluated. As shown in Fig. 2b, c, the genetic expression of both NRF1 and TFAM was significantly upregulated. Therefore, the above results confirmed that Pg treatment under optimal conditions could activate PGC-1α/NRF1/TFAM signaling, thereby inducing mitochondrial biogenesis in hMSC.

Next, we investigated whether the induction of mitochondrial biogenesis could promote the intercellular mitochondrial transfer capabilities of hMSC. It was observed that the promotion of mitochondrial biogenesis could augment mitochondrial transfer from hMSC to bleomycin (BLM)-injured TC-1 cells (induced by the treatment with BLM at 20 μg mL⁻¹) (BLM-TC-1), especially at the optimal condition of 10 μM Pg for 5 days of treatment (Supplementary Fig. S2). This finding suggested the possibility of potentiating mitochondrial transfer by increasing mitochondrial mass in donor cells. However, we also noticed that the mitochondrial transfer rate of hMSC with Pg treatment only was significantly lower than that of the Cx43-overexpressing hMSC stimulated by IONPs (Fe-hMSC), suggesting that improved mitochondrial biogenesis could only enhance the mitochondrial transfer to a certain extent. The transfer rate in this preparation was still restricted due to a limited number of gap junctions. Joint engineering applying Pg and IONPs in treatment to increase mitochondrial mass and hMSC gap junctions (Pg-Fe-hMSC) simultaneously, achieved the highest mitochondrial transfer rate (~30%), which was almost twice that of the original hMSC and was significantly higher than our previously reported Fe-hMSC (Fig. 2d). Inhibition of the expression of PGC-1α or Cx43 dramatically restricted intercellular mitochondrial transfer (Supplementary Fig. S3). These results support the need for joint engineering of hMSC to enable the optimal mitochondrial transfer capability. Furthermore, observations using transmission electron microscopy (TEM) found a typical double-membrane annular gap junction structure, which incorporates the transferred mitochondrial (Supplementary Fig. S4). This finding indicated that the observed intercellular mitochondrial transfer is probably achieved through the internalization via the gap junctions[28,29].

Negligible effects were observed on the osteogenic, adipogenic, and chondrogenic differentiation potentials of hMSC induced by joint engineering using Pg and IONPs, indicating the biocompatibility of this engineering strategy (Supplementary Fig. S5). Real-time observation of the dynamic mitochondrial transfer revealed accelerated mitochondrial transfer from Pg-Fe-hMSC to BLM-TC-1 cells compared to hMSC, in which a considerable number of mitochondria in Pg-Fe-hMSC were transferred to BLM-TC-1 cells at 2 h post-coculture; meanwhile, the number of mitochondria transferred from hMSC was much lower over the same time period (Supplementary Figs. S6 and S7, and Supplementary Movies S1 and S2). Enlarged images allow a clearer demonstration of the different mitochondrial transfer abilities between hMSC and Pg-Fe-hMSC and are shown in Supplementary Fig. S8. This rapid and efficient intercellular mitochondrial transfer capability of Pg-Fe-hMSC is critical to reorganizing mitochondrial homeostasis in MRT[30]. In particular, we previously demonstrated the capability of hMSC of injured cells-selected mitochondrial transfer, which was important to guarantee the precise delivery of mitochondria to the injured cells[23]. The present study further confirmed that this injured cells-selected

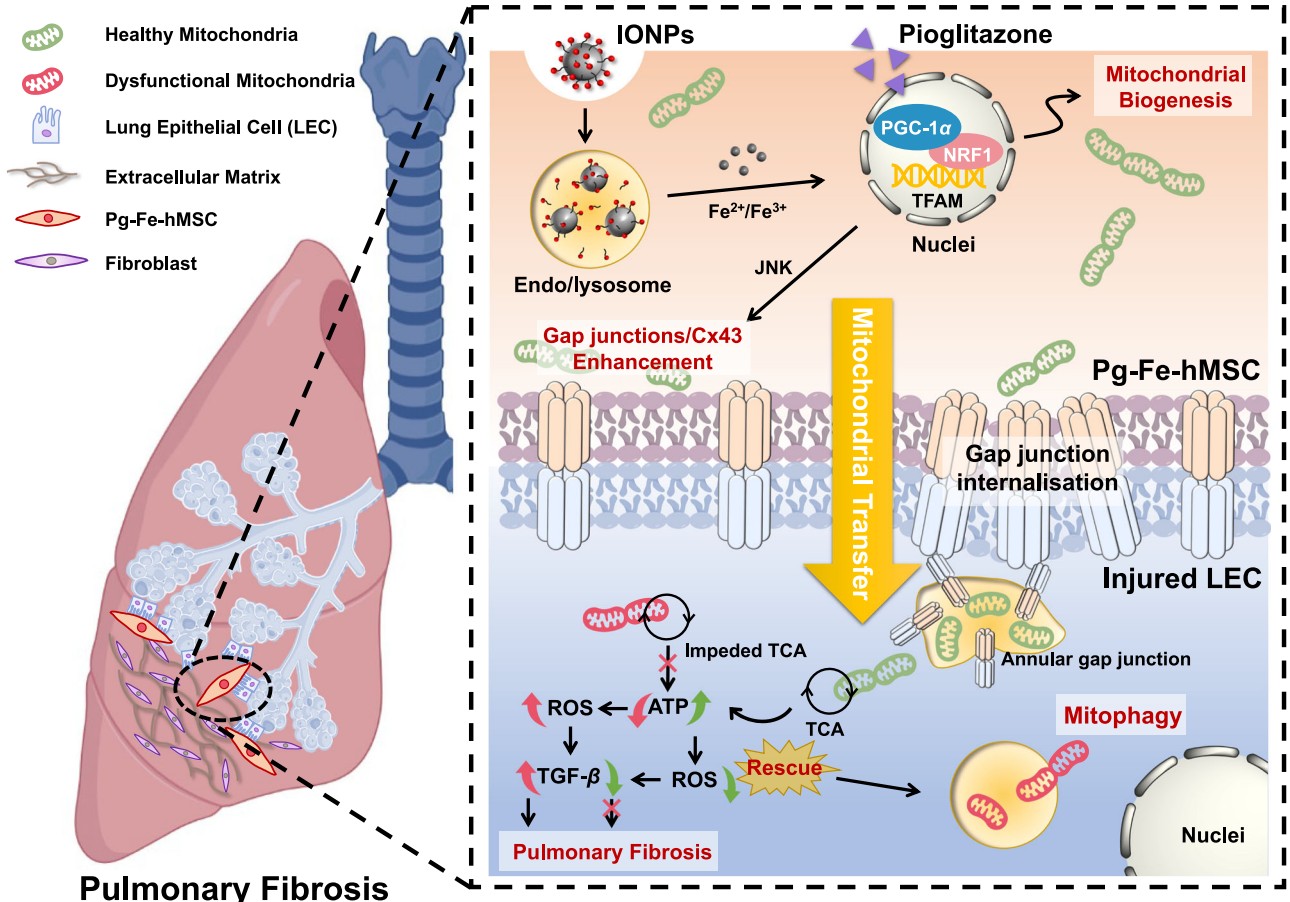

**Fig. 1 | Schematic illustration showing the strategy of using a joint-engineered mesenchymal stem cell (Pg-Fe-hMSC) to enable a powerful and sustained intercellular mitochondrial delivery specialized to injured lung epithelial cells (LECs) for the potent intervention of pulmonary fibrosis (PF).** These joint-engineered hMSC using iron oxide nanoparticles (IONPs) and pioglitazone realized powerful intercellular mitochondrial transfer, which is mainly due to the promoted mitochondrial biogenesis through the PGC-1α pathway and augmented mitochondrial transfer through the gap junction facilitated trafficking (e.g. annular gap junction mediated internalization). The Pg-Fe-hMSC can be used as both a mitochondrial generation factory and a smart vehicle for selective mitochondrial transfer to injured lung cells, showing the advantages of enabling efficient and sustained mitochondrial transfer. Consequently, Pg-Fe-hMSC achieves efficient intervention treatment for PF. This demonstrated strategy provides a potential approach to prepare mitochondria donor cells and mitochondria vehicles for workable mitochondrial replenishment treatments. Cx43 connexin 43, NRF1 nuclear respiratory factor 1, TFAM mitochondrial transcription factor A, JNK c-Jun N-terminal kinase, TCA tricarboxylic acid cycle, ROS reactive oxygen species, TGF-β transforming growth factor-β. Figure was partially created with BioRender.com.

mitochondrial transfer was not only found in injured lung epithelial cells, but was also observed in injured lung fibroblasts (Hs888Lu) and injured endothelial cells (human umbilical vein endothelial cells, HUVEC), but the transfer rate from hMSC to injured lung epithelial cell was the highest (Supplementary Fig. S9). Additionally, the treatment with Pg did not affect this important property of hMSC, which showed a high mitochondrial transfer rate towards injured TC-1 cells (Fig. 2d) and a low transfer rate with healthy TC-1 cells (Supplementary Fig. S10). Moreover, Similar to the results with lung epithelial cells, Pg-Fe-hMSC also showed the highest mitochondrial transfer rate to fibroblast and endothelial cells than hMSC, Fe-hMSC, and Pg-hMSCs, but the transfer rate was lower than that observed in lung epithelial cells (Supplementary Fig. S11).

To further verify that the enhanced mitochondrial transfer capacity of hMSC after Pg treatment was associated with mitochondrial biogenesis through the PGC-1α pathway, the relationship between the mitochondrial mass and mitochondrial transfer rate was examined. Observations using confocal laser scanning microscopy (CLSM) showed more obvious fluorescence signals of the mitochondria in Pg-hMSC than those in hMSC, indicating improved mitochondrial biogenesis of Pg-hMSC. Mitochondrial signals and mitochondrial transfer rates were both decreased when inhibiting the PGC-1α expression of

Pg-hMSC (siPGC-Pg-hMSC, Fig. 2e). The correlation between mitochondrial mass (Fig. 2f) and mitochondrial transfer rate (Fig. 2g) was further confirmed by quantitative analysis. Interestingly, genetic engineering of hMSC using *PGC-1α*-encoded plasmid demonstrated a similar up-regulation of PGC-1α, NRF1, and TFAM as treatment with Pg (Fig. 2h–j), and significantly promoted the mitochondrial transfer capacity of PGC-1α-transfected hMSC (PGC-hMSC, Fig. 2k). Therefore, we believe that treatment with Pg to induce mitochondrial biogenesis through the PGC-1α/NRF1/TFAM signaling pathway may be a potential strategy to efficiently enhance the mitochondrial transfer capacity of hMSC.

This enhanced mitochondrial biogenesis was also observed to allow sustained mitochondrial transfer. As demonstrated in Fig. 2l–o, Pg-Fe-hMSC not only achieved the highest mitochondrial transfer rate (the gating strategy is presented in Supplementary Fig. S12) but also maintained efficient mitochondrial transfer capacity for at least 96 h. Meanwhile, Fe-hMSC, which showed weak mitochondrial biogenesis, also had relatively high mitochondrial transfer rates initially but decreased rapidly after 48 h and was even lower than Pg-hMSC, probably due to mitochondrial exhaustion. This sustained mitochondrial transfer capacity of Pg-Fe-hMSC was further confirmed by microscopic observations (Supplementary Fig. S13).

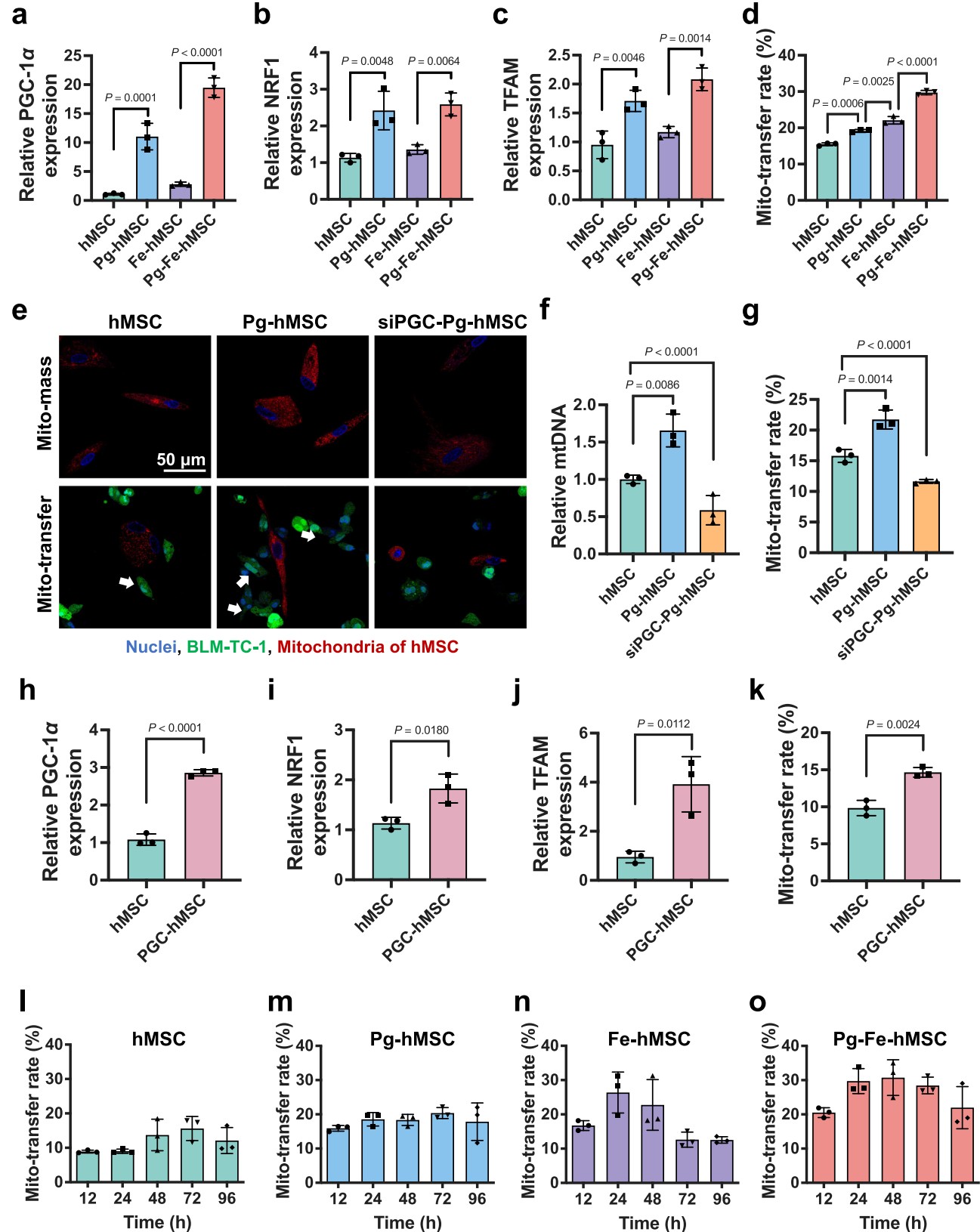

**Augmented mitochondrial biogenesis in hMSC achieved synergetic effects to improve the therapeutic potential in rescuing injured mouse LEC**

The therapeutic potential of Pg-Fe-hMSC was then examined to rescue the BLM-TC-1 cell. Compared to Pg-hMSC and Fe-hMSC, Pg-Fe-hMSC demonstrated the best therapeutic potential to restore ATP levels (Fig. 3a), decrease reactive oxygen species (ROS) levels (Fig. 3b), and restore impaired mitochondrial membrane potential (MMP) (Fig. 3c) in the BLM-TC-1 cell. As expected, Pg-Fe-hMSC also showed the best ability to protect BLM-TC-1 cells from severe death caused by BLM treatment (Fig. 3d).

**Fig. 2 | Increased mitochondrial biogenesis after pioglitazone treatment and enhanced intercellular mitochondrial transfer.** Relative **a** PGC-1 $\alpha$, **b** NRF1, and **c** TFAM expression in hMSC with different treatments ($n = 3$ biologically independent cells for each). **d** Mitochondrial transfer rates of hMSC with different treatments ($n = 3$ biologically independent cells). **e** Fluorescence images indicate the correlation between mitochondrial mass and mitochondrial transfer capacity of hMSC, Pg-hMSC, and siPGC-Pg-hMSC. Blue: nuclei; green: BLM-TC-1; red: mitochondria of hMSC; white arrows indicate transferred mitochondria. Scale bar, 50 μm. **f** Relative levels of mtDNA in hMSC, Pg-hMSC, and siPGC-Pg-hMSC ($n = 3$ biologically independent cells). **g** Quantitative analysis of mitochondrial transfer rates of hMSC, Pg-hMSC, and siPGC-Pg-hMSC ($n = 3$ biologically independent cells).

**h** Relative PGC-1α expression of hMSC before and after the gene transfection using *PGC-1α* plasmid ($n = 3$ biologically independent cells). Relative **i** NRF1 and **j** TFAM expression of hMSC before and after the gene transfection using the *PGC-1α* plasmid ($n = 3$ biologically independent cells for each). **k** Quantitative comparison of mitochondrial transfer rates between hMSC and PGC-hMSC ($n = 3$ biologically independent cells). Mitochondrial transfer rates of **l** hMSC, **m** Pg-hMSC, **n** Fe-hMSC, and **o** Pg-Fe-hMSC at different time points ($n = 3$ biologically independent cells for each). Data are presented as means ± SD. Statistical significance was analyzed using Student's *t*-test (**h–k**) or ordinary one-way analysis of variance (ANOVA) (**a–d**, **f**, **g**, **l–o**).

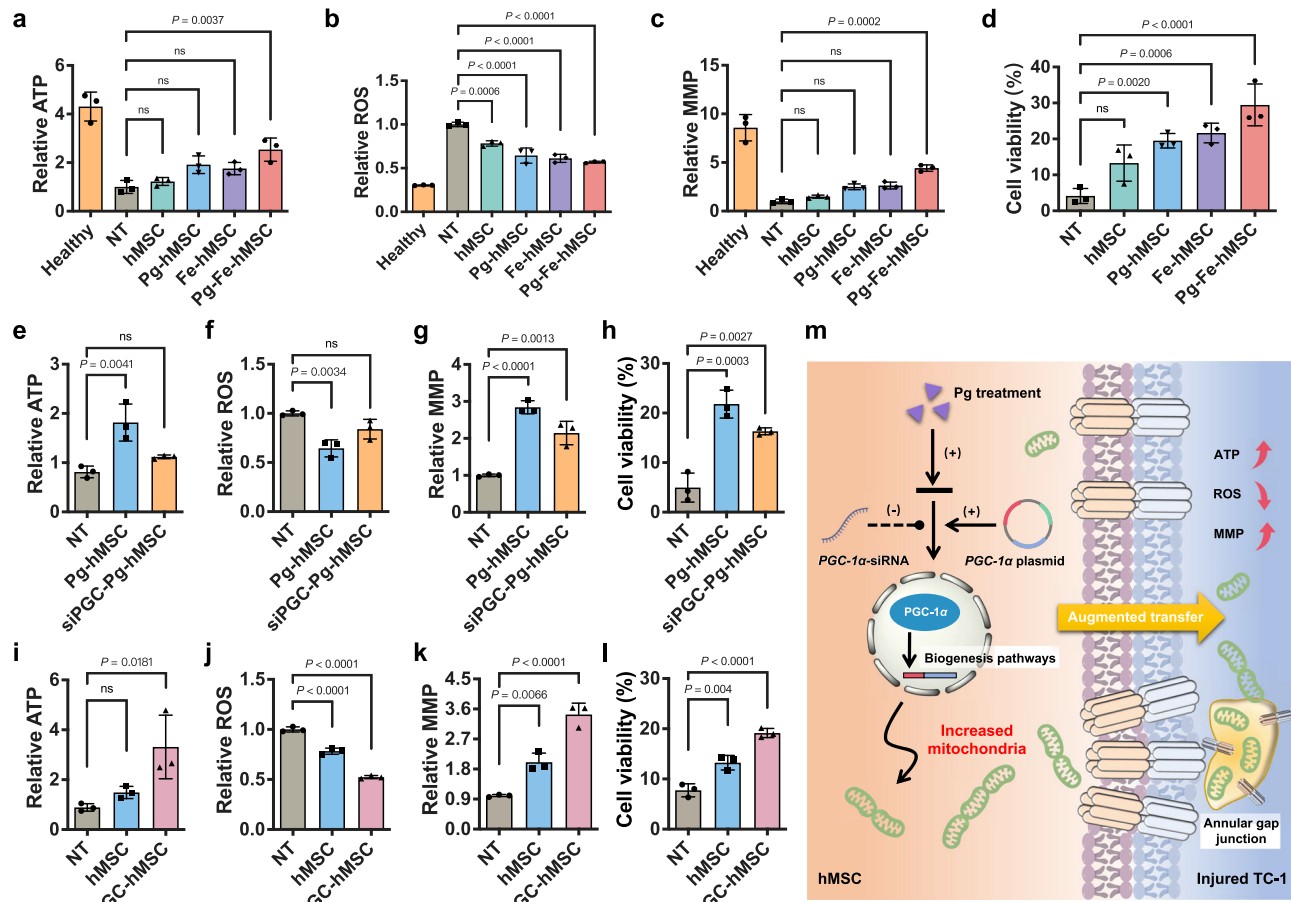

**Fig. 3 | In vitro therapeutic potential of Pg-Fe-hMSC in mouse LECs. a** ATP levels, **b** Intracellular ROS levels, **c** Mitochondrial membrane potential (MMP) levels, and **d** Cell viability of bleomycin-treated TC-1 cells (BLM-TC-1) after treatment with different engineered hMSCs ($n = 3$, biologically independent cells for each). Impacts of Pg treatment on **e** ATP levels, **f** Intracellular ROS levels, **g** MMP levels, and **h** Cell viability of BLM-TC-1 cells ($n = 3$, biologically independent cells for each).

Impacts of PGC-1α expression on **i** ATP levels, **j** Intracellular ROS levels, **k** MMP levels, and **l** Cell viability of BLM-TC-1 cells ($n = 3$, biologically independent cells for each). **m** Illustration showing the possible therapeutic mechanism of treating hMSC with Pg for augmented mitochondrial transfer via promoting the PGC-1α-dependent mitochondrial biogenesis. Data are presented as means ± SD. Statistical significance was analyzed using ordinary one-way ANOVA.

Since the critical role of IONPs in improving the intercellular mitochondrial transfer of hMSC via the gap junctions was demonstrated in our previous study[23], the impacts of promoted mitochondrial biogenesis induced by Pg treatment through the PGC-1α pathway on the therapeutical efficiency of MRT was investigated. Our results showed that the therapeutic potential of Pg-hMSC in restoring ATP levels (Fig. 3e), decreasing ROS levels (Fig. 3f), and recovering impaired MMP (Fig. 3g) in BLM-TC-1 cells was negatively affected when inhibiting PGC-1α expression. Furthermore, the protective effect of Pg-hMSC in preventing TC-1 cell apoptosis after BLM treatment was also adversely affected (Fig. 3h). Meanwhile, up-regulation of PGC-1α expression in hMSC using gene transfection demonstrated similar

effects in restoring the mitochondrial homeostasis of BLM-TC-1 cells (Fig. 3i–k), as well as protecting the TC-1 cells against the damage caused by BLM treatment (Fig. 3l). However, direct treatment of BLM-TC-1 cells with Pg showed poor therapeutic effects (Supplementary Fig. S14), which excluded any effect by Pg on the rescue of TC-1 cells. Therefore, the engineering using Pg treatment to improve the PGC-1α-dependent mitochondrial biogenesis is believed to play a positive role in enhancing the therapeutical potential of Pg-hMSC to rescue the BLM-TC-1 cells (Fig. 3m).

Moreover, different numbers of engineered hMSCs were co-cultured with injured TC-1 cells, and mitochondrial transfer rates and the corresponding therapeutic outcomes were examined. The present

study observed that cell number ratios had an impact on the mitochondrial transfer rate and largely determined the therapeutic efficiency (Supplementary Fig. S15), which suggested the importance of the mitochondrial transfer rate in MRT. In particular, the transfer rate of Pg-Fe-hMSC was as high as 40% when co-cultured with BLM-TC-1 cells in a cell number ratio of 3:1, and this is the highest intercellular mitochondrial transfer capacity reported to date, to our knowledge. These findings indicated the promising potential for inducing mitochondrial biogenesis to enhance intercellular mitochondrial transfer for an efficient MRT.

### Augmenting the mitochondrial biogenesis of Fe-hMSC achieved efficient intervention treatment of PF in the mouse model

To further confirm the potent therapeutic ability of Pg-Fe-hMSC, the classical mouse PF model established through intratracheal injection of BLM[24,31] was applied in this study. Considering the efficient therapeutic intervention of Fe-hMSC in mitigating the fibrotic progression demonstrated in our previous study[23], we delayed the first treatment time to 7 days after the initial injection of BLM in the present study, allowing the development of more severe fibrosis in the model mouse. Another reason is that TC-1 cells treated with BLM at high concentrations showed a tendency to receive more exogenous mitochondria than those treated with BLM at low concentrations (Supplementary Fig. S16). This result can probably be attributed to the severely injured TC-1 cells that require more exogenous mitochondria to restore the impaired mitochondrial homeostasis[15]. Only Pg-Fe-hMSC could satisfy this high demand for mitochondrial transfer and achieve effective therapeutic protection of TC-1 cells with severe injuries. Pg-hMSC or Fe-hMSC showed limited protection of the severely injured TC-1 cells (Supplementary Fig. S17). Therefore, we hoped to evaluate the therapeutic efficacy of different engineered hMSCs in the treatment against advanced fibrosis for a better demonstration of the therapeutic superiority of Pg-Fe-hMSC. Observations using micro-CT showed marked mitigated fibrotic progression in the lungs after treatment with Pg-Fe-hMSC, while treatment with hMSC, Pg-hMSC, and Fe-hMSC retained CT features of fibrosis (Fig. 4a). Hematoxylin and eosin (H&E) staining and Masson's trichrome staining of lung sections (harvested from three independent mice samples) further confirmed the optimal therapeutic potential of Pg-Fe-hMSC, which remarkably reduced the fibrotic areas (Fig. 4b, left panel). Furthermore, the quantitative analysis according to Masson's trichrome staining further verified the significantly relieved fibrotic ratio by Pg-Fe-hMSC (Fig. 4b, right panel). Moreover, three independent lung samples in each treatment group were homogenized to determine the collagen expression levels, showing a marked decrease in collagen deposition by Pg-Fe-hMSC (Fig. 4c). The determination of bronchoalveolar lavage fluid (BALF) collected from 10 independent lung samples further showed significantly reduced expression levels of MMP9 (Fig. 4d) and transforming growth factor-$\beta$ (TGF-$\beta$) (Fig. 4e) after the treatment with Pg-Fe-hMSC. Similarly, the measurement of hydroxyproline from eight homogenized lung samples confirmed the best therapeutic efficiency of Pg-Fe-hMSC (Fig. 4f), which was more efficient than Fe-hMSC or Pg-hMSC.

Furthermore, the mitochondrial morphologies of fibrotic lung samples were observed by TEM to assess mitochondrial damage as in previous studies[32–34]. As demonstrated in Fig. 4g, the impaired mitochondria (indicated in brown) in fibrotic lung cells showed swollen cristae structures and disrupted membranes compared to the morphology of healthy mitochondria (indicated in green). Treatment with hMSC, Pg-hMSC, or Fe-hMSC showed limited effects on the restoring of abnormal mitochondria, but Pg-Fe-hMSC successfully restored the healthy morphology of most mitochondria. Additional TEM images of the observations on a larger scale are presented in Supplementary Fig. S18. Moreover, the area and

perimeter of mitochondria were determined as previously described[33], which further supported the best therapeutic potential of Pg-Fe-hMSC to restore mitochondrial morphology in fibrotic lung cells (Fig. 4h, i). Altogether, the morphological evaluations based on the parameters mentioned above confirmed the significant therapeutic effect of Pg-Fe-hMSC in reducing the proportion of abnormal mitochondria (Fig. 4j). These results suggest that the potent therapeutic capability of Pg-Fe-hMSC against fibrosis may be closely associated with mitochondrial restoration in fibrotic lung cells.

We further compared the therapeutic efficacy of Pg-Fe-hMSC with that of Pirfenidone, one of the only two types of pharmaceutical agents approved by the Food and Drug Administration (FDA) in the clinic (the other is Nintedanib) for the treatment against PF. Interestingly, pathological observations of lung sections showed that twice Pg-Fe-hMSC treatments presented comparative therapeutic outcomes as daily administration of Pirfenidone for 21 consecutive days (Fig. 4k). In addition, quantitative calculations of the fibrotic ratio (Fig. 4l), as well as determinations of MMP9, TGF-$\beta$ and hydroxyproline levels (Fig. 4m–o) confirmed the therapeutic efficiency compared to Pirfenidone. Meanwhile, preliminary safety evaluations showed negligible side effects induced by Pg-Fe-hMSC treatment in mice (Supplementary Figs. S19 and S20).

### Mitochondrial biogenesis contributed to the efficient treatment of PF in the humanized fibrotic models

A major limitation of the mouse PF model induced by BLM is that it does not adequately mimic the irreversible progression of PF in humans[35–37]. Therefore, to further verify the impressive therapeutic potential of Pg-Fe-hMSC against PF, human lung epithelial cells (hLECs) harvested from abandoned lung tissues were applied in the following evaluations. Immunofluorescent staining with EpCAM antibodies confirmed the successful isolation of hLECs from lung tissues (Fig. 5a). Various concentrations of BLM were then applied to induce the injury of hLEC and an optimal concentration of BLM at 10 μg mL⁻¹ was selected to induce the injury (Supplementary Fig. S21). Intercellular mitochondrial transfer from different engineered hMSC to injured hLEC was examined, showing the highest mitochondrial transfer rates using Pg-Fe-hMSC (Fig. 5b). Consequently, efficient reductions in ROS production and TGF-$\beta$ secretion (Fig. 5c, d), as well as highly efficient protection of hLEC from the BLM-induced injury using Pg-Fe-hMSC was observed; meanwhile, treatments using Pg-hMSC or Fe-hMSC showed limited protection similar to that of hMSC (Fig. 5e).

A 3D multicellular human spheroid model containing human bronchial epithelial cells (BEAS-2B) and human normal lung fibroblasts (Hs888lu) was generated as a reliable human PF model according to a previous report[35] (Fig. 5f). As noted previously, BLM at a concentration of 10 μg mL⁻¹ was applied to induce fibrosis of the human lung spheroid model, which was decided based on the results of both cell viability and TGF-$\beta$ secretion (Supplementary Fig. S22). As demonstrated in Fig. 5g, the BLM treatment induced increased expressions of fibrotic markers, including $\alpha$-smooth muscle actin ($\alpha$-SMA), vimentin, and collagen type I (collagen-I), indicating the successful inducement of PF progression. While, human lung cell proliferation in the spheroid was not significantly affected, as indicated by the Ki-67 expression levels. The detailed expression levels of these markers are presented in Supplementary Fig. S23. Different engineered hMSCs were then added to this 3D multicellular human lung spheroid model after the induction of fibrosis. Pg-Fe-hMSC was observed to migrate to the inside of the spheroid after the topical administration (Supplementary Fig. S24) and achieved the highest intercellular mitochondrial transfer, whose efficiency was >30% and much higher than that of our previous Fe-hMSC (Fig. 5h). Due to the efficient mitochondrial transfer capacity, Pg-Fe-hMSC undoubtedly obtained a notable decrease in TGF-$\beta$ secretion

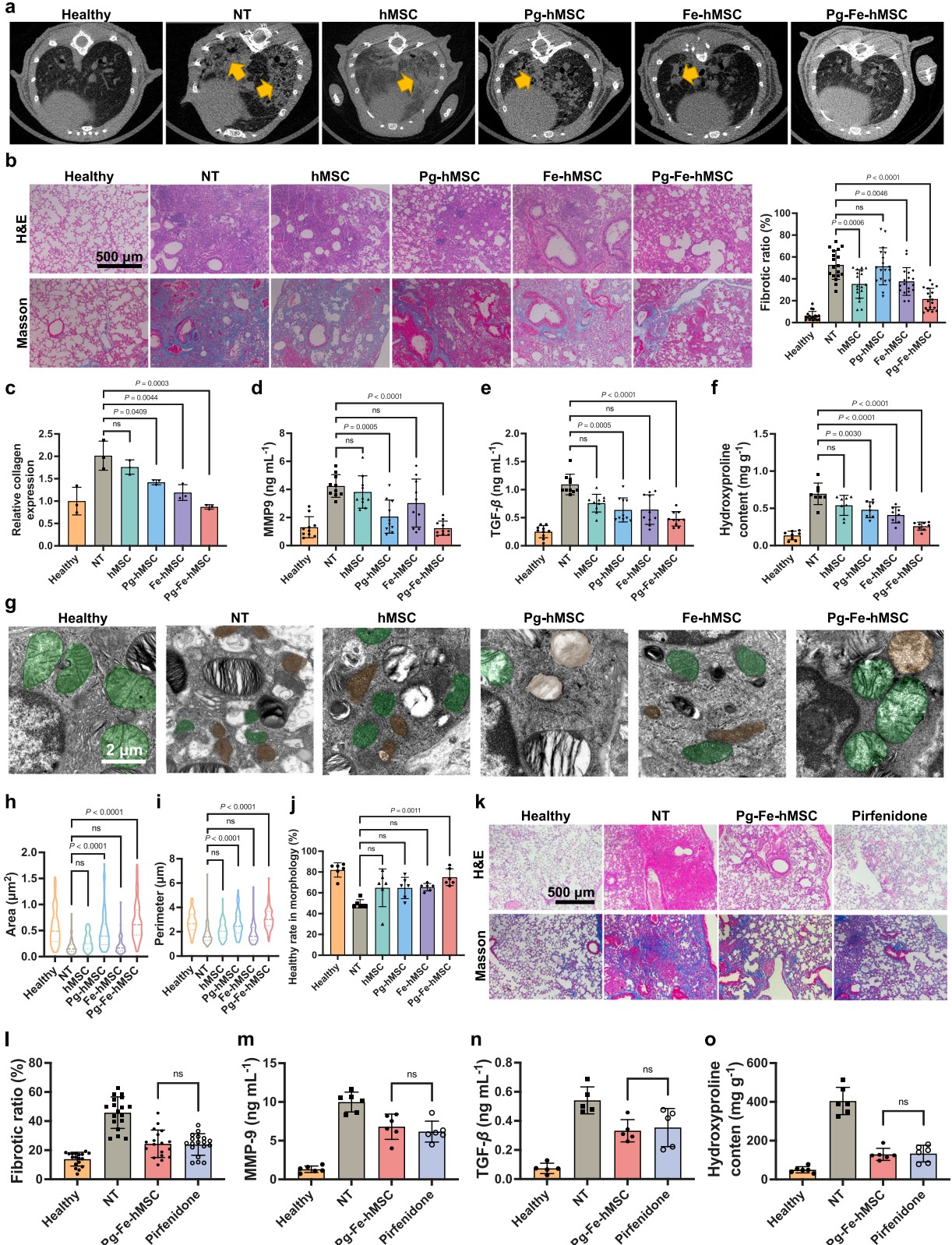

(Fig. 5i), alleviation of ROS accumulation (Fig. 5j), and reduction of cell mortality (Fig. 5k).

Overall, Pg-Fe-hMSC demonstrated significantly high intercellular mitochondrial transfer capacity, which could potentially be used against fibrosis in both monocellular and multicellular humanized fibrotic models.

## Highly efficient mitochondrial transfer restores mitochondrial homeostasis in fibrotic lung cells

Thus far, we have demonstrated the advantages of Pg-Fe-hMSC in the treatment of PF in both mouse and human fibrotic models. To further verify that this potent therapeutic potential is associated with the highly efficient mitochondrial transfer capacity of Pg-Fe-hMSC, relative

**Fig. 4 | Therapeutic potential of Pg-Fe-hMSC in the mouse PF model. a** Lung images of mice observed using micro-CT scans after treatments on Day 28 post initial administration of BLM. The yellow arrows indicate the fibrotic features. **b** Representative lung images of H&E staining and Masson's trichrome staining of mice with different treatments on Day 28, NT indicates the no treatment groups. Scale bar, 500 μm. The fibrotic ratio was calculated by measuring the percentage of blue area within the pulmonary sections ($n = 18$, six fields of view from three independent lung sections). **c** Relative collagen expression in the lung of mice after the indicated treatments ($n = 3$ biologically independent samples). Expression of **d** MMP9 and **e** TGF-$\beta$ in the bronchoalveolar fluids after the indicated treatments ($n = 10$ biologically independent samples for each). **f** Hydroxyproline content in lung homogenates after the indicated treatments ($n = 8$ biologically independent samples). **g** Observations of mitochondrial morphology using TEM. Scale bars, 2 μm. Green indicates healthy mitochondria and brown indicates impaired mitochondria. **h** Area of mitochondria ($n = 97$ in the healthy group, $n = 93$ in the NT

group, $n = 66$ in the hMSC group, $n = 79$ in the Pg-hMSC group, $n = 59$ in the Fe-hMSC group, $n = 74$ in the Pg-Fe-hMSC group, mitochondria counted from six independent lung sections for each group) measured according to TEM observations. **i** Perimeter of mitochondria calculated according to TEM observations (same replicates as **h**). **j** Ratios of healthy mitochondria according to morphological observations ($n = 6$ biologically independent samples). **k** Representative lung images following H&E staining and Masson's trichrome staining after the indicated treatments. Scale bar, 500 μm. **l** Fibrotic ratio of mice's lungs after the indicated treatments. The ratio was calculated by measuring the percentage of blue area within the pulmonary sections ($n = 18$, six fields of view from three independent lung sections). Expression of **m** MMP9, **n** TGF-$\beta$, and **o** Hydroxyproline in the bronchoalveolar fluid after the indicated treatments ($n = $ six biologically independent samples for each). Data are presented as means ± SD. Statistical significance was analyzed using ordinary one-way ANOVA.

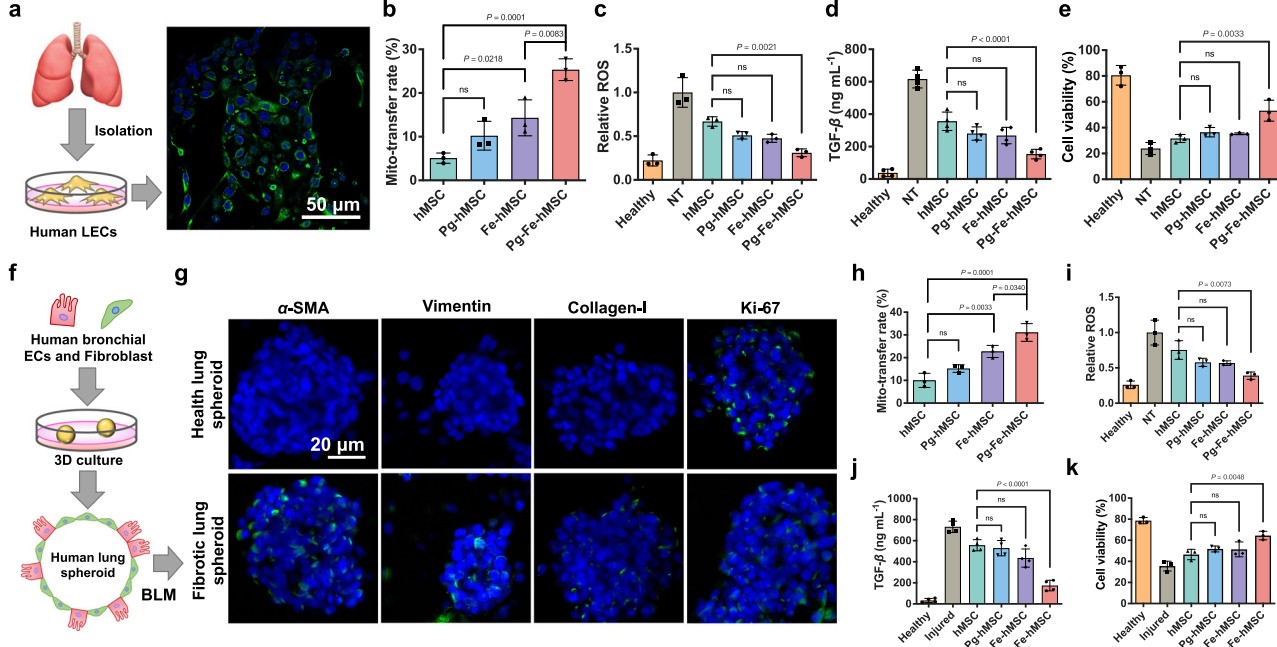

**Fig. 5 | Therapeutic potentials of Pg-Fe-hMSC in both monocellular and multicellular humanized fibrotic models. a** Schematic illustration and representative image of EpCAM immunostaining in primary human lung epithelial cells (hLECs). Scale bar, 50 μm. **b** Mitochondrial transfer rates from the indicated hMSC to the primary hLEC ($n = 3$ biologically independent samples). **c** Relative intracellular ROS levels ($n = 3$ biologically independent cells), **d** TGF-$\beta$ expression levels ($n = 4$ biologically independent cells), and **e** Viability of BLM-treated hLEC after the indicated treatment using different engineered hMSCs ($n = 3$ biologically independent cells). **f** Schematic illustration showing the preparation of the 3D multicellular human fibrotic model. **g** Representative immunostaining images showing the expression of $\alpha$-smooth muscle actin ($\alpha$-SMA), vimentin, collagen-I, and Ki-67 in the healthy

and fibrotic multicellular human spheroid models. Blue fluorescent signals indicate the cell nuclei and green signals indicate the biomarkers. Scale bar, 20 μm. **h** Mitochondrial transfer rates of different engineered hMSC in fibrotic human lung spheroids ($n = 3$ biologically independent experiments). **i** Relative intracellular ROS levels ($n = 3$ biologically independent experiments) and **j** TGF-$\beta$ expression levels of fibrotic human lung spheroids after the indicated treatment using different engineered hMSCs ($n = 4$ biologically independent experiments). **k** Viability of fibrotic human lung spheroids after the indicated treatment using different engineered hMSCs ($n = 3$ biologically independent experiments). Data are presented as means ± SD. Statistical significance was analyzed using ordinary one-way ANOVA. ECs epithelial cells.

amounts of human mitochondrial DNA (mtDNA) collected from murine lungs were quantified 24 h post the initial administration of stem cells. The highest amount of mtDNA was determined in lung tissues after systemic administration of Pg-Fe-hMSC (Fig. 6a). To further investigate the contributions of augmented intercellular mitochondrial transfer to the therapeutic efficiency of MRT, genetic analysis of messenger RNA (mRNA) was performed using mouse lung samples. The differentially expressed mitochondria-associated genes between mice treated with Fe-hMSC and Pg-Fe-hMSC were enriched and analyzed. Fifty-one genes in the mice lungs were up-regulated after treatment with Pg-Fe-hMSC (fold change > 1.2, $P < 0.05$) in comparison with Fe-hMSC (Fig. 6b). Gene ontology (GO) analysis further showed that the up-regulated genes were enriched in the mitochondrial

organization, mitophagy, the electron transport chain, and in ATP biosynthetic processes (Fig. 6c), indicating that the transferred mitochondria restored mitochondrial homeostasis in injured cells. Furthermore, the interaction networks of the up-regulated genes in the mice lungs post-treatment with Pg-Fe-hMSC were analyzed using the STRING database (https://string-db.org/), which showed changes in six critical genes including *MT-CO1*, *MT-CO2*, *MT-ND1*, *MT-ND3*, *TOMM5*, and *TOMM6* (Fig. 6d). *MT-CO1* and *MT-CO2* are genes related to ATP biosynthetic processes; *MT-ND1* and *MT-ND3* are genes related to the electron transport chain; *TOMM5* and *TOMM6* are genes responsible for mitophagy. Based on this analysis, the expressions of aforementioned genes by fibrotic lung cells were examined by polymerase chain reaction (PCR). The fibrotic lung cells showed the lowest expressions

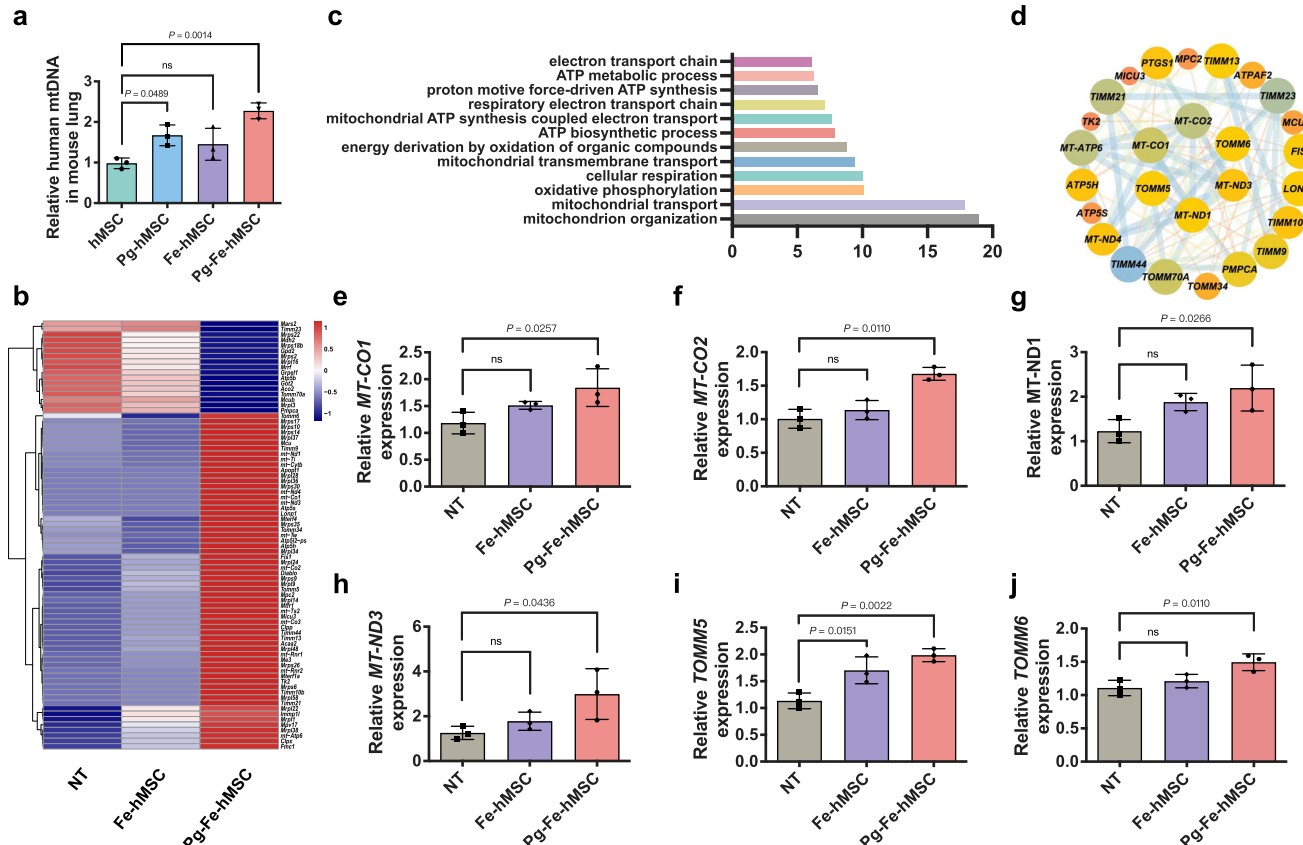

**Fig. 6 | Regulation of mitochondrial dynamics in fibrotic lung cells through highly efficient mitochondrial transfer using Pg-Fe-hMSC. a** Relative human mtDNA in mice lungs after treatment with indicated engineered hMSC (*n* = 3 biologically independent animals). **b** Heatmap of significantly differentially expressed mitochondria-related genes after the indicated treatments (*n* = 3 biologically independent animals). **c** Enriched GO pathways based on the significant differentially expressed genes between Fe-hMSC and Pg-Fe-hMSC. **d** String analysis

identified genes in PF mice treated with Pg-Fe-hMSC that were associated with mitochondrial functions. PCR analyses of differentially expressed mitochondria-related genes of **e** *MT-CO1*, **f** *MT-CO2*, **g** *MT-ND1*, **h** *MT-ND3*, **i** *TOMM5*, and **j** *TOMM6* after the indicated treatments (*n* = 3 biologically independent animals for each). Data are presented as means ± SD. Statistical significance was analyzed using ordinary one-way ANOVA.

of these genes, indicating that BLM treatment interfered with oxidative phosphorylation and ATP biosynthesis. However, treatment with Pg-Fe-hMSC significantly increased the expressions of these genes, while Fe-hMSC showed an inability to achieve comparable impacts (Fig. 6e–h). Similarly, only treatment with Pg-Fe-hMSC significantly recovered the gene expression profiles of *TOMM5* and *TOMM6* of the fibrotic lung cells (Fig. 6i, j). These findings indicated that efficient and sustained mitochondrial transfer from Pg-Fe-hMSC to fibrotic lung cells may further re-activate the mitophagy in fibrotic lung cells, thereby assisting with the restoration of mitochondrial homeostasis. This capability also partly explained the potent therapeutic potential of Pg-Fe-hMSC, because fibrotic progression will inhibit the mitophagy and result in the accumulation of misfolded protein, collagen, and extracellular matrix, which ultimately leads to irreversible scarring and pulmonary remodeling[38,39]. We also observed that the co-localization of lysosome and mitochondria decreased after treating TC-1 cells with BLM, indicating inhibited mitophagy. However, this BLM-induced inhibition of mitophagy was recovered after treatment with Pg-Fe-hMSC (Supplementary Fig. S25). The ability of Pg-Fe-hMSC to recover impaired mitophagy was then confirmed by determining the expression of LC3B I and LC3B II autophagy markers in TC-1 cells (Supplementary Fig. S26). Collectively, the above results indicated that the highly efficient transfer of exogenous mitochondria to fibrotic lung cells not only replenished the impaired mitochondria but also reversed the inhibited mitophagy, which may further contribute to the recovery of mitochondrial homeostasis and thus prevent fibrotic progression.

## Discussion
The findings of the critical roles of natural intercellular mitochondrial transfer in self-tissue repair[18,40,41] have inspired a therapeutic strategy for restoring the energy metabolism of injured cells by introducing exogenous mitochondria obtained from other cells. Currently, MSCs are considered the most promising candidate source for MRT[17,18,42,43], not only because of their inherent homing capabilities towards diseased tissues[44,45] but also because of their inherent willingness to donate mitochondria[17,22]. However, the limited efficiency of this naturally occurring mitochondrial transfer between cells currently restricts the therapeutic application of this strategy[15]. To overcome this challenge, we previously applied cellular engineering using IONPs to trigger Cx43 overexpression in hMSC, thus promoting mitochondrial transfer[23]. Nevertheless, such a strategy does not fully resolve the primary cause of restricted mitochondrial transfer, which is believed to be mainly due to the low mitochondrial contents in hMSC in their immature state[46]. Therefore, increasing the number of mitochondria in hMSC could be a potential solution to further augment the mitochondrial transfer capacity[47].

For the aforementioned reasons, the present study investigated the possibility of enhancing intercellular mitochondrial transfer by promoting mitochondrial biogenesis in hMSC, since the available number of mitochondria is closely regulated by mitochondrial biogenesis[25]. Our results demonstrated for the first time that Pg, a classical thiazolidinedione drug for type 2 diabetes treatments, can be used to efficiently promote mitochondrial biogenesis of hMSC by

activating the PGC-1α/NRF1/TFAM pathway. Additionally, no significant detrimental effects on the differentiation potentials of hMSC after Pg treatment were observed. In particular, increased mitochondrial mass in Pg-hMSCs resulted in significantly augmented mitochondrial transfer compared to those of the original hMSCs. However, they showed limited advantages compared with our previously reported Fe-hMSCs, indicating that both the number of mitochondria loaded and the mitochondrial transfer pathways determine the efficiency of mitochondrial transfer between cells. Therefore, we proposed a joint engineering strategy using Pg treatment to induce mitochondrial biogenesis, while using IONPs to up-regulate Cx43 expression, achieving a synergetic increase in the mitochondrial transfer rate. Regarding the mechanism of Cx43-facilitated mitochondrial transfer, there are still no clear conclusions. Both the Cx43 gap junction channels and the Cx43 tunneling nanotubes (TNT) have been reported to participate in intercellular mitochondrial transfer[23]. The present study observed the formation of an annular gap junction structure with mitochondria inside, indicating that the internalisation involving Cx43 gap junctions may be the mechanism responsible for the Cx43-facilitated mitochondrial transfer[28,29]. However, additional studies with more detailed data are required to improve our understanding of this interesting process. In addition to the highly efficient mitochondrial transfer capacity, Pg-Fe-hMSC also showed a sustained mitochondrial transfer at high efficiency (96 h), which was barely achievable with Fe-hMSC (48 h). Such efficient and sustained mitochondrial transfer is vital for the therapeutic potentials of MRT, which guarantees that injured cells continuously receive sufficient mitochondria to rescue their dysfunctional energy metabolism. Consequently, Pg-Fe-hMSC achieved the best therapeutic outcomes against fibrosis in both the murine model and the human cells model. Of note, the therapeutic efficacy of Pg-Fe-hMSC with two-dose administrations was even comparable with Pirfenidone treatment with multiple-dose administrations (21 consecutive days), and negligible treatment-induced side effects were observed. In the present study, we did not compare therapeutic efficacy with the other FDA-approved drug Nintedanib for PF treatment, because these two drugs share similar efficiency[48,49]. Above all, these results indicate that highly effective and sustained MRT is competitive as a promising strategy against PF progression. Furthermore, compared to the previous strategy, which applied thyroid hormone to directly promote the mitochondrial biogenesis in LECs to improve mitochondrial bioenergetics and attenuate mitochondrial-regulated apoptosis, thus inhibiting PF[24], our current MRT strategy possesses an attractive superiority due to the inherent selection of mitochondrial dysfunctional cells. Therefore, we believe that the described strategy involving Pg-Fe-hMSC has the potential for efficient and precise regulation of mitochondrial homeostasis.

In this study, we further applied bioinformatic analysis to figure out the critical role of mitochondrial transfer in the treatment against PF. Efficient and sustained mitochondrial transfer by Pg-Fe-hMSC not only restarted ATP synthesis and reduced cellular oxidative phosphorylation but also re-activated the inhibited mitophagy in injured LEC. Recovery of mitochondrial homeostasis and rebalance of energy metabolism have recently been proposed to play a critical role in the prevention of fibrotic progression[50]. Our study further provided the genetic evidence to support the claim that the transferred mitochondria rescued the injured cells and alleviated fibrotic progression by restoring mitochondrial homeostasis. Notably, the reversal of BLM-induced mitophagy inhibition through the efficient introduction of exogenous mitochondria was revealed in the present study. To our knowledge, this is a pioneering report on the use of highly efficient intercellular mitochondrial transfer to restart inhibited mitophagy. However, further studies are required to investigate additional mechanisms underlying this phenomenon.

In summary, the present study showed the promising therapeutic potential of MRT in treatment against the progression of PF in both murine and human cell models. Moreover, the engineering of MSCs with enhanced mitochondrial biogenesis and increased mitochondrial transfer capacity may provide a potential solution to overcome the major challenges of highly efficient and sustained mitochondrial transfer specifically targeting injured cells, thereby promoting the practical application of MRT in the treatment of various diseases caused by mitochondrial dysfunction.

## Methods

### Ethics declarations

All animal studies performed in this study were approved by the Animal Experimental Ethics Committee of Zhejiang University (Approval numbers: ZJU20230186) and conducted according to the committee's guidelines. The acquisitions and research applications of human lung samples were approved by the Human Research Ethics Committee of the Second Affiliated Hospital, Zhejiang University School of Medicine (Approval number: 0153). The participating patients were given informed consent under the auspices of the Institutional Review Board of the Second Affiliated Hospital, Zhejiang University School of Medicine. The handling of human mesenchymal stem cells was approved by the Medical Research Ethics Committee of Ningbo First Hospital (Approval number: 2021-R108).

### Cell culture

hMSC derived from the placenta was provided by SinoCell (Ningbo, China). All cell handling protocols were approved by the Medical Research Ethics Committee of Ningbo First Hospital. The stem cells were cultured in a complete medium composed of Human Basal Medium with Stimulatory Supplements (MesenCult™, Stemcell Technologies, Vancouver, Canada). Mouse alveolar epithelial cells (TC-1) were kindly gifted by Prof. Peihua Luo from Zhejiang University, which were purchased from Bluefcell Biotechnology (BFN60805941, Shanghai, China) and were morphologically authenticated before the study. TC-1 cells were cultured in a complete medium composed of RPMI 1640 (Zhejiang Cienry Biotechnology, Huzhou, China) containing 10% fetal bovine serum (FBS) (Wisent Biotechnology Nanjing, Nanjing, China) and 1% penicillin–streptomycin solution (Thermo Fisher Scientific, Waltham, USA). Human umbilical vein endothelial cell (HUVEC, KG419) was purchased from KeyGEN Biotech (Nanjing, China) and cultured in a complete medium composed of RPMI 1640 containing 10% FBS and 1% penicillin–streptomycin solution. Cells were authenticated by STR profiling (results of the STR analysis are demonstrated in Supplementary Table S1) before the delivering by the vendor. Human lung epithelial cells (BEAS-2B, ZQ0381) and human lung fibroblast (Hs888Lu, ZQ005) were purchased from ScienCell Research Laboratories (Shanghai, China). These two kinds of cell lines were authenticated by STR profiling before the delivery by the vendor. Results of the STR analysis can be found in Supplementary Tables S2 (BEAS-2B) and S3 (Hs888Lu). BEAS-2B cells were cultured in a Bronchial Epithelial Cell Medium (Shanghai Zhong Qiao Xin Zhou Biotechnology, Shanghai, China). Hs888Lu cells were cultured in a complete medium composed of dulbecco's modified eagle medium (DMEM) high glucose medium (Zhejiang Cienry Biotechnology, Huzhou, China), 10% FBS and 1% penicillin–streptomycin solution. All cells were cultured in the incubator (Thermo Fisher Scientific, Waltham, USA) under a humid atmosphere with 5% $CO_2$ at 37 °C.

### Preparations of different engineered hMSCs

hMSCs at a certain concentration were seeded and cultured overnight, followed by the indicated procedures listed below to obtain Pg-hMSCs, Fe-hMSCs, and Pg-Fe-hMSCs, respectively. Briefly, hMSCs were co-cultured with a Pg solution at the optimal concentration of 10 μM for 5 consecutive days to obtain the Pg-hMSCs. The preparation of Fe-hMSCs or Fe-Pg-hMSCs was performed by the treatment with IONPs[23]. In brief, hMSCs or Pg-hMSCs were incubated with IONPs (30 μg mL⁻¹) in serum-

free medium for 1 h, followed by replacement with fresh complete medium for an additional 24 h to acquire the Fe-hMSC or Pg-Fe-hMSC.

## Preparations of fibrotic model cells

BLM purchased from Shanghai Aladdin Bio-Chem Technology (Shanghai, China) was applied to induce the injury of diverse lung cell lines, including TC-1, Hs888Lu, and BEAS-2B cells, and the injury of HUVEC. Briefly, BLM at indicated concentration (20 μg mL$^{-1}$ for TC-1 and 10 μg mL$^{-1}$ for other kinds of cells) was added into the culture medium and incubated with cells at 37 °C with 5% $CO_2$ for 24 h.

## Assessments of mitochondrial transfer in vitro

The mitochondrial transfer capacity of the engineered hMSC was determined by both observation using a CLSM (Leica, Wetzlar, Germany), and quantitative analysis using a flow cytometer (BD Biosciences, San Jose, USA). In short, different engineered hMSCs were stained by MitoTracker® probes (MitoTracker Deep Red or Mito-Tracker CMXRos, Invitrogen Life Technologies, Carlsbad, USA) for 30 min, and then washed three times with neutral phosphate buffer (PBS). The mitochondria-labeled hMSC was co-cultured with a green fluorescent protein (GFP)-tagged TC-1 cells (GFP-TC-1) for 24 h at different cell number ratios. GFP-TC-1 cells were prepared through lentiviral transfection according to the manufacturer's instructions (Jikai Gene, Shanghai, China). For the observation of mitochondrial transfer, the co-cultured cells were fixed by using 4% paraformaldehyde and stained with DAPI. The detected fluorescence signals of mitochondria localized in GFP-TC-1 were considered as transferred mitochondria. To quantify the mitochondrial transfer rate, the co-cultured cells were detached and collected by trypsin. Cells with mitochondrial fluorescence signals and GFP-TC-1 were gated with a flow cytometer to quantify the mitochondrial transfer rate.

## Dynamic observation of mitochondrial transfer using real-time CLSM

Pg-Fe-hMSC and hMSC were co-cultured with GFP tagged BLM-TC-1, and their mitochondrial transfer ability was monitored using real-time CLSM observation (Leica LAS X, Leica, Wetzlar, Germany) started at 2 h post the co-culture. The co-culture system was incubated in a humid atmosphere of 5% $CO_2$ and 37 °C. Fluorescent images were automatically taken every 5 min and processed into short movies using Image J software (ij152-win-java8, National Institutes of Health, Bethesda, USA).

## Comparison of mitochondrial transfer rates toward different cells

To compare the mitochondrial transfer capacity of hMSC to different cell types, HUEVC and Hs888Lu cells were applied to co-culture with hMSC. These two kinds of cells were treated with BLM at 20 μg mL$^{-1}$ to induce injury, followed by the addition of different engineered hMSC. After 24 h co-culture, the mitochondrial transfer rates were determined by flow cytometer (BD Biosciences, San Jose, USA). In addition, HUEVC and Hs888Lu cells without BLM treatment were also applied as comparisons to confirm the injured cells selected transfer of mitochondria.

## Isolation of GFP-TC-1 cells from the co-culture system

GFP-TC-1 cells were isolated by a fluorescence-activated sorting flow cytometer (Beckman, Brea, USA). Briefly, the co-cultured cells were collected by trypsinization and suspended in neutral PBS. The mixed cells were first gated, and GFP-positive cells were then gated and collected.

## Mitochondrial ROS and mitochondrial MMP determination

MitoSOX red mitochondrial superoxide indicator (Yeasen Biotechnology, Shanghai, China) was applied to detect mitochondrial ROS

of BLM-TC-1. Briefly, the indicator was diluted to 5 μM in Hanks' balanced salt solution (HBSS) and added to the cell samples. After staining at 37 °C for 10 min in a dark environment, the stained cells were washed and suspended in HBSS. The cell suspensions were assessed by flow cytometry, and the positive cells were quantitatively analyzed to determine the ROS. For the evaluation of MMP, tetra-methylrhodamine, methyl ester, and perchlorate [TMRM] (Yuanye Biotechnology, Shanghai, China) were applied. Briefly, the reagent was diluted to 100 nM in a serum-free medium and added to the cell samples. After staining at 37 °C for 10 min in a dark environment, the stained cells were washed and suspended in PBS. The collected cell suspensions were assessed by flow cytometry, and the mean fluorescence intensity of TMRM was tested for quantifying MMP.

## Evaluations of multilineage differentiation potential of Pg-Fe-hMSC

To exclude any possible influences on the differentiation potential of hMSC, the osteogenic, adipogenic, and chondrogenic differentiation potentials of Pg-Fe-hMSC were analyzed by the mesenchymal stem cell osteogenic differentiation kit, adipogenic differentiation kit, and chondrogenic differentiation kit (CHEM Biotechnology, Shanghai, China), respectively. Briefly, the cells were seeded on six-well plates and induced with the corresponding differentiation medium for 2 weeks before staining. The osteogenic differentiation was evaluated by detecting the alkaline phosphatase (ALP) activity using the staining of calcium phosphate-cobalt sulfide. The adipogenic differentiation was evaluated by observing lipid vacuoles within the cytoplasm after oil-red O staining. The chondrogenic differentiation was evaluated by staining the produced glycosaminoglycans (GAG) using alcian blue.

## Determination of cell viability

Alexa Fluor 647 conjugated Annexin V and PI apoptosis detection kit (Yeasen Biotechnology, Shanghai, China) were applied to determine the viability of TC-1 cells after different treatments. In brief, cells were harvested by trypsinization using EDTA-free trypsin (Gibco BRL, Gaithersburg, USA) and subsequently washed for two times using pre-cooled PBS. The collected cells were then resuspended in 1× binding buffer (100 μL) and incubated with annexin V (5 μL) and PI staining solutions (10 μL) at room temperature for 15 min in a dark environment. Afterward, 100 μL 1× binding buffer was added to dilute the stained cells, followed by flow cytometry analysis immediately. The viability was evaluated by counting the double-negative-stained cells in the gated population.

## Measurement of intracellular ATP levels

The intracellular ATP levels were measured with an ATP assay kit (Beyotime Biotechnology, Shanghai, China) according to the manufacturer's instructions. Typically, the adherent cells or tissue samples were lysed by lysis buffer, and centrifuged at 12,000 × $g$ under 4 °C for 5 min. The supernatants were collected for photochemical reaction with ATP working solutions. The light units of samples and ATP standard solutions were obtained by a luminometer (GloMax, Promega, Madison, USA). The ATP levels were calculated according to the calibration curve made by ATP standard solutions. To exclude the influence of the difference in cell amounts on ATP measurements, the amounts of proteins in each sample were determined using a BCA protein assay reagent kit (Beyotime Biotechnology, Shanghai, China). The intracellular ATP levels were calculated as ATP concentration per cell protein and normalized by the control group.

## siRNA transfection

The expressions of Cx43 and PGC-1α by hMSC were down-regulated by the gene transfection using the corresponding siR-NAs. The sequence of *Cx43-siRNA* and *PGC-1α-siRNA* (produced by

Sangong Biotechnology, Shanghai, China) are as follows: *Cx43-siRNA* (Sense: AGGAAGAGAAACUGAACAATT, Antisense: UUGUUC AGUUUCUCUUCCUTT), *PGC-1α-siRNA* (sense: GACUAUUGCCAG UCAAUUAAUTT, antisense: AUUAAUUGACUGGCAAUAGUCTT). LipoHigh liposomes were applied for the siRNA transfection according to the manufacturer's instructions. In brief, diluted siRNA and LipoHigh liposomes at indicated concentrations were mixed and incubated for 15 min to prepare the gene complexes. The gene complexes were then added to hMSC in a serum-free medium for 3 h, after which the culture system was replaced with a complete medium. To verify the downregulation efficiency, western blot analysis was applied.

### Gene transfection of *PGC-1α* plasmid
Briefly, hMSC was seeded on a 24-well plate overnight. When the cell convergence reached 70–90%, the culture medium was discarded and replaced with fresh medium without serum. *PGC-1 α* plasmid was diluted to a proper concentration in a serum-free medium, mixing with LipoHigh liposome for preparing the gene complexes. The gene complexes were then added to hMSC in a serum-free medium for 4 h, after which the culture system was replaced with a complete medium. To verify the gene transfection efficiency, western blot analysis was applied.

### Establishment of mouse PF model
The mouse PF model was established through intratracheal injection of BLM, which has been widely accepted as a classical animal model of PF. In brief, C57BL/6 mice (male, 6-week-old) purchased from Shanghai SLAC Laboratory Animal Co. Ltd (Shanghai, China) were housed in an animal feeding system of individually ventilated cages (IVC system, ZJ-4, Suzhou Fengshi Laboratory Equipment Co., Ltd., Suzhou, China) under standard laboratory housing conditions ($25 \pm 1\,°C$, 50% relative humidity and 12 h/12 h dark/light cycle) for 1 week. Afterward, C57BL/6 mice were anesthetized and intratracheally administered BLM at the concentration of $0.8\,\text{g}\,\text{kg}^{-1}$ in 50 μL PBS to induce the PF. All animal studies performed in this study were approved by the Animal Experimental Ethics Committee of Zhejiang University and conducted according to the committee's guidelines.

### Antifibrosis treatment in mouse PF model
The different engineered hMSCs ($5 \times 10^5$ cells per mouse) were administered intravenously to PF mice two times on Day 7 and Day 10 after the initial challenge with BLM, respectively. Meanwhile, Pirfenidone (Shanghai Aladdin Biochemical Technology, Shanghai, China) was administered daily to PF mice via intragastric delivery from Days 7 to 28 after the initial BLM challenge as a positive control. After 28 days of the initial BLM challenge, the progression of PF was first examined by micro-CT. After the examinations, both lung and BALF samples were collected from PF mice for further evaluation. H&E staining and Masson's trichrome staining of lung samples were performed for histological evaluation. Furthermore, the levels of hydroxyproline, ATP, and collagen-I in lung samples were determined. RNA sequencing of the collected lung samples was also performed. Meanwhile, quantitative analyses of the expression of TGF-$\beta$ and MMP9 in BALF samples were performed.

### Micro-CT scanning
The micro-CT scanner (U-CT-XUHR, Milabs, Houten, The Netherlands) was applied to observe the fibrotic progression of lungs in a mouse model after the treatments. Mice were anaesthetized by halothane during the scanning. The parameters of the scanning were as follows: the X-ray system of the scanner was set to an accurate mode, with a voltage of 50 kV, a current of 0.21 mA, an exposure time of 75 ms, and a total scan time of 5–10 min.

### BALF collection
BALF was collected for the assessment of TGF-$\beta$ and MMP9 levels in lung tissues. In brief, mice were sacrificed at the indicated time point, followed by the separation and cannula of the trachea. The left lung was clamped with hemostatic forceps and then flushed with 1 mL saline for collection of lung lavage. The collected BALF was stored on ice before subsequent determinations.

### Enzyme-linked immunosorbent assay (Elisa) of MMP9 and TGF-$\beta$
The concentrations of MMP9 and TGF-$\beta$ in BALF were measured by Elisa kits (Boster, Wuhan, China) according to the manufacturer's protocol. Briefly, the collected BALF was added to the precoated 96-well plates and incubated at 37 °C for 90 min. Then the solutions in plates were discarded, followed by incubation with MMP9 or TGF-$\beta$ antibodies at 37 °C for 60 min. The incubated plates were washed with washing buffer thrice, after which avidin–peroxidase complex was added for another incubation at 37 °C for 30 min. Subsequently, the plates were washed with washing buffer five times, and 3,3',5,5'-tetramethyl benaidine (TMB substrate) were introduced for chromogenic reactions at 37 °C for 20 min in a dark environment. Consequently, the reactions were terminated, and the optical density (OD) value of the samples was measured using a microplate reader (ELX800, BioTek Instrument, Vermont, USA) at 450 nm. The MMP9 and TGF-$\beta$ levels were calculated according to the calibration curves, which were established using standard solutions.

### Histological assessments of lung sections
Lung samples of mice under different treatments were harvested at indicated time points, fixed in a 4% formaldehyde solution (Boster, Wuhan, China), and subsequently embedded in paraffin. The lung samples were then cut into 6 μm-thick sections, followed by H&E and Masson's trichrome staining. The images of lung sections were taken by a Panoramic MIDI slide scanner (3DHISTECH Kft., Budapest, Hungary). The collagen depositions of the lung were evaluated by quantifying the ratio of blue area in the Masson's trichrome section using Image J software.

### Hydroxyproline determination
Hydroxyproline detection kit (Solarbio, Beijing, China) was used for the determination of hydroxyproline. Briefly, lung samples were harvested and cut into small pieces, followed by dissolving by acid hydrolysis at 110 °C for 6 h. After cooling to room temperature, the solutions were centrifugated at 16,000 rpm for 20 min to collect supernatants. Afterward, certain amounts of sodium hydroxide were added to the supernatants for neutralization. A series of reaction agents were subsequently added for the chromogenic reactions following the manufacturer's instructions and the OD values were measured at 560 nm.

### RNA sequencing analysis
Total RNA was extracted from fresh mice lungs ($n = 3$ biologically independent animals) at the treatment endpoint (28 days post the initial BLM challenge), followed by qualification of the purity and integrity using Agilent 2100 bioanalyzer. After a series of procedures of mRNA enrichment, cDNA synthesis, and library quality assessment, RNA sequencing was performed using the Illumina sequencing system by Novogene (Tianjin, China). Gene expression levels were then measured in terms of transcripts per kilobase per million. The bioinformatic analysis was conducted with the aid of online databases and platforms (https://string-db.org/, https://metascape.org/).

### TEM observation of mitochondria
Cell or tissue samples were collected and fixed with 4% pre-cooled glutaraldehyde (Sigma-Aldrich Chemical, St. Louis, USA) overnight at 4 °C, followed by PBS washing on the next day. A series of treatments

were subsequently applied, including 1% osmic acid fixing, water washing, 2% uranyl acetate staining, gradient concentration ethanol, and acetone dehydrating. Consequently, the dehydrated samples were embedded and sliced into 100 nm samples. The morphologies of mitochondria were observed by TEM (Tecnai G2 spirit, Thermo Fisher Scientific, Waltham, USA). Mitochondria with healthy morphology were counted and the area and perimeter of mitochondria were quantified by Image J software (six independent lung sections for each group).

### Isolation and quantification of RNA via qRT-PCR
Total RNA was extracted from cell or lung tissues by subsequent treatments with TRIzol, ethanol, centrifugation, and washing. The purity of RNA was determined by ultraviolet spectrophotometer (NanoDrop One, Thermo Fisher Scientific, Waltham, USA) and electrophoresis before reverse transcription. Then, the reverse transcription system containing total RNAs, SuperScript™III First-Strand Synthesis SuperMix (Thermo Fisher Scientific, Waltham, USA), enzyme mix, and RNase-free water was constructed for obtaining cDNA, followed by real-time PCR. The primer sequences for PCR can be found in Supplementary Table S4.

### Human mtDNA measurements by PCR
The relative human mitochondrial copy numbers of the cell samples were determined by PCR analysis. The primers of human-specific mtDNA applied were as follows: Forward 5′-CCCCACAAACCCCATT ACTAAACCCA-3′, Reverse 5′-TTTCATCATGCGGAGA TGTTGGATGG-3′, and human nuclear β globin DNA (nDNA) were: Forward 5′-GAAGAG CCAAGGACAGGTAC-3′, Reverse 5′-GGAAAATAGACCAATAGGCAG-3′. The relative copy numbers of human mtDNA were quantified through $2^{-(\Delta\Delta Ct)}$ by real-time PCR analysis (StepOne PULS, ABI, Foster, USA).

### Isolation and culture of hLECs
Human lung samples were harvested from the removed lung cancer tissues. The participating patients (two male patients and two female patients, aged from 18 to 60 years old. The sex of patients was not taken into consideration, and only the discarded lung tissues after tumor surgical resection were collected) were provided informed consent under the auspices of the Institutional Review Board of the Second Affiliated Hospital of the Zhejiang University School of Medicine. Non-cancerous specimens were carefully isolated, washed with sterile PBS solution, and cut into 1 mm × 1 mm pieces. Prewarmed PBS solutions containing collagenase I (2 mg mL$^{-1}$) and deoxyribonuclease I (50 U mL$^{-1}$) were applied to incubate and digest the tissues for 30 min at a temperature of 37 °C. After incubation for 30 min, the digested lung tissues were disassociated by vigorous shaking, followed by a filter through 40 μm filters. The filtered cell suspensions were washed with cold PBS and centrifuged to collect the cell precipitates. The purified cells were then stained with CD31 (Biolegend, 102405, 1:200), CD45 (Biolegend, 103132, 1:200), and CD326 (Biolegend, 118205, 1:200) antibodies in a dark environment at 4 °C for 30 min. The stained cells were washed and resuspended in cold PBS. hLECs were sorted using a flow cytometer according to the surface marker of CD31$^-$CD45$^-$CD326$^+$. The collected hLECs were cultured in a complete DMEM/F12 medium (Thermo Fisher Scientific, Waltham, USA). Immunofluorescent staining with EpCAM antibodies (Affinity, DF6311, 1:500) was applied to confirm the successful isolation of hLECs from lung tissues.

### Preparation of humanized 3D fibrotic lung spheroid
The 3D fibrotic lung spheroid was prepared based on a previously reported method[35] with several modifications. Briefly, BEAS-2B human bronchial epithelial cells and Hs888lu human normal lung fibroblasts were mixed in a 2:1 ratio of cell number. Matrigel (Yeasen Biotechnology, Shanghai, China) was thawed at 4 °C and then added to the

prepared cell suspensions. The Matrigel-containing cell suspensions were then removed from the cell culture plates at a density of 200 μL per cm$^2$, followed by gelation at 37 °C for 30 min. After the solidification of Matrigel, a complete cell culture medium was supplied for the nutritional support of the lung spheroid. To induce the fibrotic phenotype of the lung spheroid, BLM treatment (10 μg mL$^{-1}$) was performed on Day 7 after the cell mixture and for 48 h. The expression levels of α-SMA, collagen-I, vimentin, and Ki-67 were then determined by immunofluorescent staining to confirm the fibrosis of the humanized 3D lung spheroid. In brief, the collected samples (three biologically independent samples for each group) were firstly labeled with the primary antibodies of α-SMA (Boster, BM3902, 1:100), collagen-I (Proteintech, 14695-1-AP, 1:500), vimentin (Boster, BM4029, 1:100) and Ki-67 (Boster, PB9026, 1:200), followed by the incubation with the secondary antibody of FITC labeled goat anti-rabbit IgG (H + L) (Boster, BA1105, 1:100) for 1 h. The fluorescence signals were observed by CLSM and the density of fluorescence signals was quantified using Image J software.

### Determination of mitochondrial transfer rates and therapeutic effects on 3D fibrotic lung spheroid
Different engineered hMSCs were added from the top to the 3D fibrotic lung spheroid at the cell number ratio of 1:3 (hMSC:lung cell) after induction of fibrosis. The distribution of Pg-Fe-hMSCs was observed using a CLSM (IX81-FV1000, Olympus, Tokyo, Japan). Pg-Fe-hMSCs were indicated by the mitochondrial staining using Mito-Tracker® probes (MitoTracker Deep Red) as described above. BEAS-2B and Hs888lu cells were stained using the cell membrane probe of DiO (DiOC$_{18}$(3)) (Yeasen Biotechnology, Shanghai, China). The acquired original images were reconstructed by the Imaris software (version 9.3.1, Bitplane AG, Zurich, Switzerland). The mitochondrial transfer rate and the therapeutic efficiency were quantified at 24 h post the co-culture, which the determination methods were similar to the determination with TC-1 cells described above.

### Preliminary safety evaluations of Pg-Fe-hMSC in mouse
To evaluate the possible influences of the treatment using Pg-Fe-hMSC on mouse, Pg-Fe-hMSC at the dosage of $5 \times 10^5$ cells per mouse were intravenously administered to healthy mice twice on Day 1 and Day 4, respectively. Healthy mice without cell infusion (PBS treatment) were taken as a control group. On Day 21, blood samples and major organs including heart, lung, liver, kidney, and spleen were collected. The levels of white blood cells (WBC), alanine transaminase (ALT), and aspartate transaminase (AST) were quantified and compared between the Pg-Fe-hMSC group and the control group. In addition, H&E staining was performed on the harvested major organs to examine the possible side effects.

### Determination of the expressions of LC3B II/I
Relative expressions of LC3B II/I were assessed by western blot analyses. In brief, the proteins in collected samples were extracted by protein extraction reagent containing a halt protease inhibitor cocktail (Thermo Fisher Scientific, Waltham, USA), which were further quantified by a BCA protein concentration determination kit (Beyotime Biotechnology, Shanghai, China). Separation gels and concentration gels were prepared, and indicated proteins were added for electrophoresis. The proteins on gels were transferred to polyvinylidene fluoride (PVDF) membranes (Millipore, Bedford, USA). After transferring for 2 h, the PVDF membranes were blocked, followed by 5 min washing thrice. The primary antibodies of LC3B (CST, 43566, 1:1000) and GAPDH (Abcam, ab181602, 1:10000) were diluted to certain concentrations and added to the above-washed membranes. After overnight incubation at 4 °C, the membranes were washed and incubated with goat anti-rabbit IgG (H + L) (Thermo Pierce, 31210, 1:5000) or goat anti-mouse IgG (H + L) (Thermo Pierce, 31431, 1:5000) for 1 h at room

temperature, followed by repeated washing. The antibody information is listed in Supplementary Table S5. Finally, the bands on the membranes were visualized by incubating with an enhanced chemiluminescence kit (Thermo Fisher Scientific, Waltham, USA). The OD values of visualized bands were quantified by Image J software. The expressions of indicated proteins were normalized by the level of GAPDH.

### Statistical analysis

All data produced were analyzed by GraphPad Prism version 8.2.1 (GraphPad Software; www.graphpad.com). The experiments performed in triplicate or over three times were expressed as mean ± standard deviation (SD). For comparison of differences between the two groups, a two-sided Student's *t*-test was performed. For determining differences among multiple groups, ordinary one-way analysis of variance (ANOVA) was applied.

### Reporting summary

Further information on research design is available in the Nature Portfolio Reporting Summary linked to this article.

## Data availability

All data needed to evaluate the conclusions in the paper are present in the paper and/or the Supplementary Information. Any additional requests for information can be directed to, and will be fulfilled by, the corresponding authors. The RNA sequencing data of pulmonary fibrosis model mice generated in this study have been deposited in the GEO database under accession code "GSE228129". The raw data generated in this study are provided in the Source Data file. The original figures used in this study are available in the "Figshare database [https://doi.org/10.6084/m9.figshare.23684148]". Source data are provided with this paper.

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

## Acknowledgements
We would like to thank Prof. Peihua Luo from the Institute of Pharmacology and Toxicology, College of Pharmaceutical Sciences, Zhejiang University for the gifted mouse TC-1 cells. We thank the Center of Cryo-Electron Microscopy (CCEM), Zhejiang University for technical assistance on TEM, and Ms. Yingying Huang (Core Facilities, School of Medicine, Zhejiang University) for her assistance with cell sorting. We also thank Ms. Shuangshuang Liu from the Core Facilities, Zhejiang University School of Medicine for her technical assistance on CLSM. This work was supported by the National Natural Science Foundation of China no. U22A20383 (J.G.), Natural Science Foundation of Zhejiang Province no. LD22H300002 (J.G.), and the Fundamental Research Funds for the Central Universities no. 226-2022-00125 (T.Z.).

## Author contributions
J.G. and T.Z. conceived the project and designed the experiments. T.Z. and T.H. carried out the experiments and collected the data. R.L., Y.S., and N.L. assisted in the cell experiments. R.L., H.S., X.Z., and J.Z. assisted in the animal studies. J.S. instructed the animal experiments. X.L. and X.F. analyzed the RNA-sequencing. D.X., B.Z., and L.H. conducted the isolation of human lung epithelial cells. T.Z. and T.H. wrote the original manuscript. X.J. polished the figures and assisted in data analysis. J.G. revised the manuscript. All authors discussed the results and commented on the manuscript.

## Competing interests
The authors declare no competing interests.
