## [Peer Review File · Nature Communications]

REVIEWER COMMENTS

Reviewer #1 (Remarks to the Author):

Study by Huang et al focuses on evaluation on the efficiency of manipulation of MSC by simultaneous overexpression of Connexin43 and PGC-1a to improve efficiency of mitochondrial transfer from MSCs to target cells. Authors state that mitochondrial transfer occurs via gap junctions and evaluate therapeutic efficacy of modified MSCs to alleviate lung fibrosis. I have several important questions regarding this study.

Most crucially there is a conceptual problem with the mechanism of mitochondrial transfer via Connexin 43 Gap-junctions.

According to the Sosinsky GE, Nicholson BJ. Structural organization of gap junction channels. *Biochim Biophys Acta*. 2005 Jun 10;1711(2):99-125. doi: 10.1016/j.bbamem.2005.04.001.

“The channel pore narrows from an apparent ~ 40 Å diameter at the cytoplasmic opening to ~ 15 Å at the extracellular side of the bilayer before widening to ~ 25 Å in the extracellular region.

However, it is expected that the diameters should be 10 Å less due to the contributions of the side chains that were not resolved at 7 Å resolution. The narrowest part of the channel would be only 5 Å in diameter. Cx43 channels are permeable to molecules of at least 15 Å.”

Diameter of mitochondrion is appr. 1000 nanometers (2000 Å) – which makes it physically impossible for mitochondrion to pass through gap junctions.

The first publication which sparked interest in the participation of gap junctions in the MSC mitochondrial transfer (Islam et al., *Nature Medicine*, 2012) did not provide any direct evidence that mitochondria are actually passing through the channels, it rather implied that chemical communication via channels (e.g. Ca signalling) facilitated mitochondria uptake via EVs and nanotubules.

In the subsequent publications by Li et al, *Theranostics*, 2019 and Yao et al., *Stem Cell Reports*, 2018, including previous publication by authors of the present manuscript in *Sci Adv*, 2021, no convincing evidence was presented that mitochondria are directly passing through gap junctions.

It is absolutely fundamental that authors should provide solid evidence (e.g. good electron microscopy imaging or other high resolution imaging) showing that mitochondria are found within gap junction between MSCs and their target cells – that will be a revolution in our understanding of intercellular communication. Gap junctions may be also facilitating mitochondria transfer indirectly via chemical signalling which would perhaps change phosphorylation status of the cytoskeletal proteins etc., but this should be clearly demonstrated before making statements regarding role of Con43 in mitochondrial transfer.

Figure 4 – in vivo model of lung fibrosis – It is widely accepted that after IV administration MSCs are trapped in the lung vasculature for 24-48 hrs and eventually are being cleared by immune cells, this clearance contributes to the reprogramming of immune cells towards anti-inflammatory phenotypes and is considered to be important if not the main mechanism of MSC effect. Following the logic of the present manuscript, to prove the concept of importance of direct modulation of bleomycin activated cells by mitochondrial transplantation, MSC should be able to actively infiltrate into the lung parenchyma and communicate with resident mesenchymal and epithelial cells to transfer their mitochondria. This also raises another question of the ratio between MSC/target cells, how many target cells 1 donor MSC cell should be able to modulate to achieve therapeutic effect. Authors should provide clear evidence of MSCs localization (and duration of their presence) in the lung tissue, with the evidence of mitochondrial transfer in vivo.

Figure 5 – authors refer to the organoid model as spheroid, however there is no lumen inside (according to the images), it looks like a cell aggregate. Not clear what physiological relevance this model has and how well it recapitulates features of the human IPF. It is not clear how MSCs were added to this aggregate – after induction of fibrosis or simultaneously, were they incorporated into the spheroid or migrated inside from the topical administration?

Reviewer #2 (Remarks to the Author):

This paper by Huang and colleagues describes a mechanism of mitochondrial transfer from engineered human mesenchymal stem cells (hMSCs) to diseased cells as potential therapeutic strategy for pulmonary fibrosis (PF). The authors used a combination of in vitro, in vivo and ex vivo approaches to show that engineered hMSCs, treated with pioglitazone to increase the mitochondrial content, transfer mitochondria to recipient cells and have anti-fibrotic effect in the lung.

Overall, the paper is not carefully written, and this limits the enthusiasm for the paper. The title is difficult to interpret and does not catch the attention of the readers.

The abstract is almost totally focused on the mitochondrial transfer mechanism. Although this is good, because this method is potentially applicable to other diseases with mitochondrial alterations, a more clear link with PF should be provided. Why PF? Why pioglitazone? Any links with PGC1a dysfunction during PF? Why the comparison with pirfenidone? These questions should be addressed in the abstract and throughout the manuscript to connect the 2 topics of this paper. The introduction is lengthy, an idea could be to summarize background, significance and main results and move all the additional thoughts in the discussion section.

In vivo methods to study PF are adequate, however it is not indicated why some readouts (such as collagen expression) have fewer biological replicates than the others.

The entire paper is focused on epithelial cells. Although it is known that alveolar epithelial cells present mitochondrial alterations, other cell types in the fibrotic lungs, in particular pathogenic fibroblasts (see Caporarello et al. Thorax 2019) manifest reduced expression of PGC1a and mitochondrial dysfunction, but this is not mentioned. What the authors think about mitochondrial uptake by other cell types? The engineered cells are injected intravenously, what about vascular endothelial cells? It would be interesting to further address which type is more prone to the mitochondrial uptake and the respective contributions to the anti-fibrotic effect.

An important paper in the field by Yu and colleagues (Nature Medicine 2018) was cited to support the in vivo model of PF, but it was not discussed. In this paper, the Authors demonstrated that the thyroid hormone has an anti-fibrotic effect by promoting mitochondrial biogenesis, improving mitochondrial bioenergetics and attenuating apoptosis in alveolar epithelial cells both in vitro and in vivo (in a bleomycin model of PF) through PGC1a. What are the thoughts of the Authors in this regard? How the injection of engineered hMSCs and the transfer of mitochondria may have a better translational value compared to using a drug that mimic the same effects?

Reviewer #3 (Remarks to the Author):

Summary:

The article by Huang et al. is a follow-up to their previous study published in 2021. In the previous study, they evaluated the role of iron oxide nanoparticles to augment gap junctions for improving mitochondrial transfer. In this study, they tested the hypothesis of increasing mitochondrial biogenesis to improve mitochondria replacement therapy (MRT); transferring mitochondria from a donor cell such as hMSC (human mesenchymal stem cell) to an injured lung cell. In this study, they first used a bleomycin-injured mouse TC-1 alveolar epithelial cell line to show proof of MRT in vitro. Pioglitazone (Pg) was used to induce biogenesis successfully, and when combined with iron oxide particles, they observed a synergistic effect with significantly improved mitochondrial transfer (Pg-Fe-hMSCs). They go on to show that Pg biogenesis works through the PGC-1 α /NRF1/TFAM pathway, using siRNA and plasmid, for knockdown and upregulation experiments respectively.

Having successfully shown mechanistic effects in an in vitro model using mouse alveolar epithelial cells (AEC), they move on to testing their joint-engineered therapy (Pe-Fe-hMSCs) in a rodent mouse model of pulmonary fibrosis. Once again, they see similar results in the rodent model, where the joint-engineered therapy performed better compared to individual-engineered therapies in preventing fibrotic lung disease progression. Significantly, in this study, lungs treated with joint-engineered therapy show (i) better morphology (ii) lower collagen accumulation, and (iii) reduced fibrotic markers (i.e., TGF). Then they show similar results in fibrosis models developed using human airway cells and spheroids. Finally, they show that their treatment can reverse mitophagy inhibition induced by bleomycin.

Together, the authors present a novel alternative treatment modality for pulmonary fibrotic lung disease.

Revisions:

Significant revisions need to be made to this article for it to be published. My comments are below.

L64 – “.....spontaneous mitochondrial donation from the normal to dysfunctional cells.”. Dysfunctional cells could be injured or genetically abnormal. Please explain your choice of words here. Perhaps it is better to say injured here, as the entire article is about mitochondria replenishment therapy for treating bleomycin-induced cell injury.

L83 – “...transfer mitochondria selectively to the dysfunctional cells that require more energy.” Liu et al. and Gomzikova et al., both state that healthy cells can transfer mitochondria to each other during physiologic conditions. Do you think Pg-hMSCs could increase mitochondrial transfer to healthy cells? Is there a threshold beyond which cells will transfer their mitochondria? Have you done any experiments looking at the transfer of mitochondria between healthy cells?

Liu, D., Gao, Y., Liu, J. et al. Intercellular mitochondrial transfer as a means of tissue revitalization. *Sig Transduct Target Ther* 6, 65 (2021). <https://doi.org/10.1038/s41392-020-00440-z>

Gomzikova MO, James V, Rizvanov AA. Mitochondria Donation by Mesenchymal Stem Cells: Current Understanding and Mitochondria Transplantation Strategies. *Front Cell Dev Biol.* 2021 Apr 7;9:653322.

L 98 – “demobnstrated” should be demonstrated. Typo

L 110 – “we thereby naming it as high powered MSC”. Incorrect grammar. Should be – we thereby named it as high powered MSC – or – we are thereby naming it as high powered MSC.

L114 – Especially for a disease like Idiopathic Pulmonary Fibrosis (IPF) where there are only 2 FDA-approved drugs, and when both drugs are available for research through vendors

(e.g., selleckchem), why didn't the authors try Nintedanib? What is the rationale for choosing Pirfenidone over Nintedanib for this study?

L116 – Use either artificial or engineered to talk about the modified MSCs. For example, “....the present study provides a promising strategy of using engineered hMSC to obtain mitochondrial...”. Unsure if using both adds more information. If you feel otherwise, please justify usage.

L117 - Hence, the present study provides a promising strategy of using artificial (“artificial..”; typo) engineered hMSC to obtain (“the” should be deleted) mitochondrial carrier cells with high mitochondrial biogenesis and efficient intercellular mitochondrial delivery capabilities, which can be used as a powerful mitochondrial donor cells and vehicles for treating mitochondrial dysfunctional diseases.

Typos are mentioned in parentheses. The underlined statement is superfluous. You already mention that these cells are mitochondrial carrier cells, and they have efficient delivery capabilities. No need to mention again that they are powerful donor cells and vehicles.

L123 – joint-engineered should be joint-engineered. Typo

L125 – specilaised should be “specialized”

L131 – “...with autonomous navigation ability to target mitochondrial dysfunctional AEC after systemic administration...”. My question is whether the AECs are specifically targeted? It doesn’t appear to be so. This sentence is misleading.

L146 - Supplementary Figure 1 – You need to mention whether MFI is the median or mean fluorescence index. In the figure caption, you do say that it is mean +/- SD. So, it does appear to be the mean. However, the expansion of the acronym MFI should be mentioned somewhere within the manuscript or the supplementary document. After a quick search, I did not find this.

L146 – Regarding Supplementary Figure 1 – It would help to have a label within the graph artwork that says (a) 1-day treatment, and (b) 5-day treatment

L195 – The Figure 2e panels could use color channel legends displayed within the figure, as there are multiple color channels. While this information is listed in the figure caption, it would be useful for the reader to be able to see this information within the image.

L206 – for panel figures 2I to 2o the authors should include the cell type for each of the graphs within the image. This would help to make side-by-side comparisons. This information is listed in the figure caption, but it should also be included within the image itself.

L184 – The videos and the still images of the videos in the form of supplementary figures 5 and 6 are great. While I can see red mitochondria in BLM-TC-1 cells, it is not very clear. Could the authors comment if there is a better way to show the transfer of mitochondria between two cells using live imaging?

L243-254 – This comment corresponds to image panels 3e to 3h and panels 3i to 3l. In the former series (3e to 3h), PGC-1 α is inhibited/downregulated, and based on the supplementary data, this is using siRNA. In the latter series (3i to 3l), PHG-1 α is upregulated using gene transfection. Correspondingly for the latter series of images, the graph's x-axis labels read "PGC-1 α -hMSC". This makes sense. For the former series of images, it is not clear why the authors decided to use the x-axis label "Pg-shMSC"? Does this signify treatment with si-RNA? The use of this specific x-axis label is confusing, and the reason behind its usage is unclear. It seems like the label should have been "siPGC-1 α -Pg-hMSC". Please explain. The same naming convention is also used for the image panel in Figure 2e.

L264 – This comment is regarding supplementary figure 10 – please add a legend to figure panels (a to d; e to h; i to l). It would help to have the cell-type legends in the figure.

L297 – "treating" should be treated. Typo

L309 – This comment pertains to Figure 4a. Why does the CT of the lung for the hMSC group look translucent as opposed to other groups? Please explain.

L313 – why are statistical analyses and comparisons in Figure panels 4c-f made with the NT group? However, in Figure panels, 3a-d statistical analyses and comparisons are made with the hMSC group and not the NT group. Please explain.

L324 – The transmission electron microscopy image comparisons do not accurately show what is conveyed by the authors. That is, it does not show that Pg-Fe-hMSCs successfully reduced the proportion of abnormal mitochondria. Adding supplementary figure 13 is appreciated, but it still doesn't seem to show the described pattern.

L377 – The lung images of Pg-Fe-hMSC in figures 4k and 4b look very different. The former seems to have almost normal airspace as opposed to the latter. Also, based on Masson's staining there seems to be high variability between these two experiments. Please comment on the level of variability observed when treated mice using Pg-Fe-hMSCs.

L365 – “assistes” should be assists. typo

L368 – The authors claim that they have harvested human (alveolar epithelial cells) AECs from abandoned lung tissues using EpCAM antibodies. Unfortunately, this is incorrect and misleading. Alveolar epithelial cells refer to pure populations of cells of either type I or type II or type I&II cells. Based on the technique used for isolation, the authors have isolated small airway epithelial cells (SAEC), and these can include bronchiolar epithelial cells, basal stem cells, and alveolar epithelial cells (type I and II). It is incorrect to call these alveolar epithelial cells, especially when these cells have not been characterized. Please modify the manuscript to reflect these changes. Alternatively, I am happy to receive justification from your end regarding the naming scheme of AECs. In Supplementary Figure 16, the same cells are called human lung epithelial cells. Please also make sure to have consistent nomenclature in both the manuscript and supplementary materials.

L388 – the use of cell lines to create a PF model is commendable. However, in this case, the results presented by the authors are semi-quantitative at best. Immunofluorescence staining shows α -SMA, vimentin, collagen type I, and ki-67. Quantitative results for these markers are missing. Without this information, the authors cannot make a conclusive statement that they have induced PF phenotype in these cells. Please provide additional data.

L399 – “monocelluar” should be monocellular.

L481 – In this first line of the discussion, it is unclear if the authors are talking about intracellular (within cell transfer) or intercellular (between cells) transfer of mitochondria. I believe this should have been intercellular. Please explain your choice of term.

Point-by-point Response to the Reviewers' Comments

Remarks to the Author:

Reviewer #1:

Study by Huang et al focuses on evaluation on the efficiency of manipulation of MSC by simultaneous overexpression of Connexin43 and PGC-1a to improve efficiency of mitochondrial transfer from MSCs to target cells. Authors state that mitochondrial transfer occurs via gap junctions and evaluate therapeutic efficacy of modified MSCs to alleviate lung fibrosis. I have several important questions regarding this study.

Response: We greatly appreciate the reviewer's valuable comments. Several modifications have been made in the revised manuscript, especially the comments regarding to the mitochondrial transfer through gap junctions, in response to those helpful suggestions.

1. Most crucially there is a conceptual problem with the mechanism of mitochondrial transfer via Connexin 43 Gap-junctions. According to the Sosinsky GE, Nicholson BJ. Structural organization of gap junction channels. *Biochim Biophys Acta.* 2005 Jun 10;1711(2):99-125. doi: 10.1016/j.bbamem.2005.04.001. "The channel pore narrows from an apparent ~40 Å diameter at the cytoplasmic opening to ~15 Å at the extracellular side of the bilayer before widening to ~25 Å in the extracellular region. However, it is expected that the diameters should be 10 Å less due to the contributions of the side chains that were not resolved at 7 Å resolution. The narrowest part of the channel would be only 5 Å in diameter. Cx43 channels are permeable to molecules of at least 15 Å." Diameter of mitochondrion is appr. 1000 nanometers (2000 Å) – which makes it physically impossible for mitochondrion to pass through gap junctions. The first publication which sparked interest in the participation of gap junctions in the MSC mitochondrial transfer (Islam et al., *Nature Medicine*, 2012) did not provide any direct evidence that mitochondria are actually passing through the channels, it rather implied that chemical communication via channels (e.g. Ca signalling) facilitated mitochondria uptake via EVs and nanotubules. In the subsequent publications by Li et al, *Theranostics*, 2019 and Yao et al., *Stem Cell Reports*, 2018, including

previous publication by authors of the present manuscript in Sci Adv, 2021, no convincing evidence was presented that mitochondria are directly passing through gap junctions. It is absolutely fundamental that authors should provide solid evidence (e.g. good electron microscopy imaging or other high resolution imaging) showing that mitochondria are found within gap junction between MSCs and their target cells – that will be a revolution in our understanding of intercellular communication. Gap junctions may be also facilitating mitochondria transfer indirectly via chemical signalling which would perhaps change phosphorylation status of the cytoskeletal proteins etc., but this should be clearly demonstrated before making statements regarding role of Con43 in mitochondrial transfer.

Response: Thank you for this insightful suggestion. As mentioned by the reviewer, several previous studies had pointed out the critical role of Cx43 in mediating the intercellular transfer of mitochondria. Inspired by their results, we proposed the idea of promoting intercellular mitochondrial transfer through the upregulation of Cx43 protein using IONPs. We then observed the significant impacts of gap junction protein Cx43 on the mitochondrial transfer between cells in our previous study. However, we further found that such ability was restricted during the treatment of seriously injured cells, which partly results from the fast exhaustion of mitochondria in MSC. Therefore, in the present study, we focused on investigating the promotion of mitochondrial biogenesis to further augment the mitochondrial transfer rate and sustained transfer ability.

We think the reviewer proposed an important question regarding the roles of Cx43 gap junctions in mitochondrial transfer. Therefore, we performed a detailed investigation with additional experiments to assess this critical issue. In fact, as early as 1982, Forbes et al. observed the frequent association between gap junctions and mitochondria in mammalian myocardial sections (*Tissue Cell. 1982;14(1):25-37*). The authors observed that the gap junction of length 153.8 pm had 66.3 pm of its length (43.1%) to be closely juxtaposed to mitochondria. Interestingly, mitochondrion-gap junction “sandwiches” were observed, where mitochondria in different cells were aligned with one another across an interval formed by gap junction membranes (Fig. R1).

Fig. R1. An extensive gap junction is juxtaposed to three mitochondria, two of which (M) are in different cells, yet face one another and thus sandwich the gap junction. Magnification $\times 86,000$. (*Tissue Cell*. 1982;14(1):25-37).

In addition to the “sandwich” gap junction, the authors also found circular gap junctions in which mitochondria were encircled (Fig. R2). The authors proposed that these circular gap junctions are continuous with the extracellular fluid space and are derived from intrusions from one cell into another.

Fig. R2. Enveloped mitochondria in a circular gap junction. Magnification $\times 130,000$. (*Tissue Cell*. 1982;14(1):25-37).

The following studies, as mentioned by the reviewer, further confirmed that the gap junction channels participate in the intercellular mitochondrial transfer

(*Nat Med.* 2012; 18(5): 759-65, *Theranostics.* 2019;9(7):2017-35). Our previous study also showed that the transferred mitochondria are co-localised with the Cx43 gap junctions (Fig. R3) (*Sci Adv.* 2021;7(40):eabj0534). Nevertheless, these studies only show that the Cx43 gap junctions are essential in mediating the mitochondrial transfer; how the mitochondria transfer occurs *via* gap junction channels has not been demonstrated, as pointed out by the reviewer.

Fig. R3. Images of Cx43 expression and its relationship with the intercellular transfer of mitochondria. Red arrows indicate Cx43 expression localised in the transferred mitochondria; white arrows indicate the transferred mitochondria. Scale bars, 20 μm . (*Sci Adv.* 2021;7(40):eabj0534).

To the best of our knowledge, few studies have described the transfer process of mitochondria *via* gap junction channels. Norris et al. applied three-dimensional electron microscopy to observe the incorporation of mitochondria into annular gap junctions. The authors showed that whole organelles, including mitochondria and endosomes, are incorporated into double-membrane vesicles called connexosomes or annular gap junctions, resulting in a gap junction internalisation (*Traffic.* 2021;22(6):174-179). As shown in Fig. R4, mitochondria were present in cellular protrusions that connect to nearby cells *via* gap junctions. The incorporated mitochondria were then fully enclosed within internalised gap junctions of the receiving cell.

Fig. R4. Invaginated gap junctions protruding into neighbouring cells surround organelles including mitochondria. (*Traffic*. 2021;22(6):174-179).

This gap junction internalisation is proposed to structurally resemble trogocytosis, in which part of a cell is engulfed by a contacting cell. Such a mechanism may also explain our previous finding that cell-to-cell contact is essential for an efficient intercellular mitochondrial transfer (*Sci Adv*. 2021;7(40):eabj0534). In another study, mitochondria were observed to be shed through the formation of large protrusions to adjacent cells (*Proc Natl Acad Sci U S A*. 2014;111(26):9633-8), (Fig. R5).

Fig. R5. Mitochondria with an intact cristae structure (arrows) are interspersed with irregular membranous bodies (*Proc Natl Acad Sci U S A*. 2014;111(26):9633-8).

Apart from the mitochondrial transfer, James W Smyth and Robin M Shaw had summarised the life cycle of Cx43 gap junctions (*Heart Rhythm*.

2012;9(1):151-3). Connexons can be endocytosed by juxtaposed cells through the formation of double-membrane annular gap junction structures and be degraded in the “receptor” cells (Fig. R6).

Fig. R6. Schematic representation of the life cycle of connexin 43 gap junctions. Connexin 43 hemichannels are packaged into vesicles at the trans-Golgi network from where they traffic along microtubules that can use specific plus-end binding proteins and adherens junctions to directly localise the delivery of connexons to the intercalated disc. Once in the plasma membrane, connexons bind to hemichannels of juxtaposed cells and form gap junctions. Connexons are endocytosed and degraded, and gap junctions are internalised as double-membrane “annular” gap junction structures destined for degradation. (*Heart Rhythm*. 2012;9(1):151-3).

Therefore, considering these works and our previous findings, we consider the expression “gap junction internalisation” more precise to describe the intercellular mitochondria transfer *via* gap junctions.

To confirm this postulation, we observed gap junction internalisation between Pg-Fe-hMSC and TC-1 cells using transmission electron microscopy (TEM). Despite the difficulty of observing the dynamic transfer process using TEM, incorporated mitochondria were detected in irregular membranous bodies as previously reported (Supplementary Fig. S4). This suggests that the mitochondria of hMSC may be internalised by injured TC-1 cells in annular gap junctions and then released after gap junction degradation.

Supplementary Fig. S4. TEM images of incorporated mitochondria in an annular gap junction. A typical double-membrane annular gap junction structure with incorporated mitochondria is observed after the mitochondrial transfer. Scale bar, 500 nm.

Furthermore, as we previously summarised, several different mechanisms are involved in the intercellular mitochondrial transfer (*J Control Release. 2022;343:89-106*). In addition to the Cx43 gap junction channels, Cx43 tunnelling nanotubes (TNT) are another frequently reported mechanism for intercellular mitochondria transfer, partly because they are more conveniently observed *in vitro* than the gap junctions. It is postulated that gap junctions connect the end of a nanotube to another cell, and thus mitochondria can be transferred *via* a nanotube (*Stem Cells. 2016;34(8):2210-23*). We also observed such mechanism between Fe-hMSC and bleomycin-treated TC-1 cells (BLM-TC-1) (Fig. R7).

Fig. R7. Intercellular mitochondrial transfer *via* Cx43 tunnelling nanotubes (TNT). Blue: cell

nuclei; green: Cx43 protein; yellow: mitochondria. Scale bar: 20 μm .

To conclude, we think the reviewer provided an important suggestion to improve our manuscript. Studies that uncover the detailed mechanisms of the gap junction-facilitated mitochondrial transfer will provide a vital advance in the field. Considering that the present study is more focused on augmenting gap junction-facilitated mitochondrial transfer *via* the promotion of mitochondrial biogenesis, we would like to add the TEM results as supplementary material (Supplementary Fig. S4) to support the gap junction internalisation as a potential mechanism. We also modified the Schematic illustration and corresponding descriptions (Fig. 1) for a clearer demonstration of the gap junction-mediated mitochondrial transfer. In addition, the different potential mechanisms of mitochondrial transfer are now discussed in the revised manuscript.

Revision:

● **Fig. 1**

Fig. 1. Schematic illustration showing the strategy of using a joint-engineered mesenchymal stem cell (Pg-Fe-hMSC) to enable a powerful and sustained intercellular mitochondrial delivery specialised to injured lung epithelial cells (LECs)

for the potent intervention of pulmonary fibrosis (PF). These joint-engineered hMSC using iron oxide nanoparticles (IONPs) and pioglitazone realised powerful intercellular mitochondrial transfer, which is mainly due to the promoted mitochondrial biogenesis through the PGC-1 α pathway and augmented mitochondrial transfer through the gap junction facilitated trafficking (e.g. annular gap junction mediated internalisation). The Pg-Fe-hMSC can be used as both a mitochondrial generation factory and a smart vehicle for selective mitochondrial transfer to injured lung cells, showing the advantages of enabling efficient and sustained mitochondrial transfer. Consequently, Pg-Fe-hMSC achieves efficient intervention treatment for PF. This demonstrated strategy provides a potential approach to prepare mitochondria donor cells and mitochondria vehicles for workable mitochondrial replenishment treatments.

- **Results, Line 163**

“Furthermore, observations using transmission electron microscopy (TEM) found a typical double-membrane annular gap junction structure, which incorporates the transferred mitochondrial (Supplementary Fig. S4). This finding indicated that the observed intercellular mitochondrial transfer is probably achieved through the internalisation *via* the gap junctions^{28,29}.”

- **Supplementary Fig. S4**

Supplementary Fig. S4. TEM images of incorporated mitochondria in an annular gap junction. A typical double-membrane annular gap junction structure with incorporated mitochondria is observed after the mitochondrial transfer. Scale bar, 500 nm.

● **Discussion, Line 517**

“Regarding to the mechanism of Cx43 facilitated mitochondrial transfer, there are still no clear conclusions. Both the Cx43 gap junction channels and the Cx43 tunnelling nanotubes (TNT) have been reported to participate in intercellular mitochondrial transfer²³. The present study observed the formation of an annular gap junction structure with mitochondria inside, indicating that the internalisation involving Cx43 gap junctions may be the mechanism responsible for the Cx43-facilitated mitochondrial transfer^{28,29}. However, additional studies with more detailed data are required to improve our understandings of this interesting process.”

References

23. Huang, T., *et al.* Iron oxide nanoparticles augment the intercellular mitochondrial transfer-mediated therapy. *Sci Adv* **7**, eabj0534 (2021).
28. Norris, R.P. Transfer of mitochondria and endosomes between cells by gap junction internalization. *Traffic* **22**, 174-179 (2021).
29. Smyth, J.W. & Shaw, R.M. The gap junction life cycle. *Heart Rhythm* **9**, 151-153 (2012).

2. Figure 4 – in vivo model of lung fibrosis – It is widely accepted that after IV administration MSCs are trapped in the lung vasculature for 24-48 hrs and eventually are being cleared by immune cells, this clearance contributes to the reprogramming of immune cells towards anti-inflammatory phenotypes and is considered to be important if not the main mechanism of MSC effect. Following the logic of the present manuscript, to prove the concept of importance of direct modulation of bleomycin activated cells by mitochondrial transplantation, MSC should be able to actively infiltrate into the lung parenchyma and communicate with resident mesenchymal and epithelial cells to transfer their mitochondria. This also raises another question of the ratio between MSC/target cells, how many target cells 1 donor MSC cell should be able to modulate to achieve therapeutic effect. Authors should provide clear evidence of MSCs localization

(and duration of their presence) in the lung tissue, with the evidence of mitochondrial transfer *in vivo*.

Response: Thank you for your important question and suggestion. We have previously demonstrated the lung-targeting ability of MSCs toward the fibrotic lung in a mouse model, and that MSCs reside in fibrotic lungs for more than 10 days (*Sci Adv.* 2021;7(40):eabj0534) (Fig. R8).

Fig. R8. *in vivo* distribution of MSCs at indicated time points. (*Sci Adv.* 2021;7(40):eabj0534).

In addition, as suggested by the reviewer, we also previously investigated whether the systemically injected hMSCs infiltrated into the lung parenchyma and communicated with lung epithelial cells. As presented in Fig. R9, iron-labelled hMSCs co-localised with lung epithelial cells. Moreover, the *in vivo* mitochondrial transfer from hMSCs to lung epithelial cells was also observed (Fig. R10A), and the transfer efficiency was further quantitatively determined by detecting copies of human mitochondrial DNA in mouse epithelial cells (Fig. R10B).

Fig. R9. Localisation of engineered MSCs in pulmonary sections on day 10. The upper panel shows immunofluorescent staining. Blue: nuclei; green: lung epithelial cells; red: mitochondria of engineered MSCs. The lower panel shows Prussian blue staining. Black arrows indicate the localisation of engineered MSCs. Scale bars, 50 µm. (*Sci Adv.* 2021;7(40):eabj0534).

Fig. R10. Representative images of recipient mouse pulmonary sections after immunofluorescence staining. The transferred mitochondria originating from MSCs are indicated by white arrows. Blue: nuclei; green: MSCs; red: mitochondria of MSCs; purple: lung epithelial cells. Scale bars, 10 µm (left) and 5 µm (right). (*Sci Adv.* 2021;7(40):eabj0534).

Therefore, our previous studies has showed the targeting capability of MSCs toward lung epithelial cells, as well as the mitochondrial transfer *in vivo*.

Furthermore, the reviewer asked an important question regarding the suitable cell number ratio between MSCs and lung epithelial cells. We also recognised this important issue. However, it is difficult to determine the cell number ratio *in vivo*, partly because of our current technical inability to precisely count the number of MSCs in the lungs. Therefore, we evaluated therapeutic efficiency using different cell number ratios *in vitro*. As showed in Supplementary Fig. S15, hMSCs were co-cultured with BLM-treated TC-1 cells at cell number ratios from 1:3 to 3:1 based on previous studies (*BMC Cell Biol.* 2010;11:29. *Am J Respir Cell Mol Biol.* 2014;51(3):455-65. *Stem Cell Res Ther.* 2016;7(1):91. *Stem Cell Reports.* 2016;7(4):749-763). The results demonstrated that higher ratios of hMSCs achieved a stronger protective effect. For our Pg-Fe-hMSCs, the therapeutic effect was observed even at the low cell number ratio of 1:3 (Pg-Fe-hMSCs:BLM-TC-1).

Supplementary Fig. S15. Cell number ratios impact the mitochondrial transfer rates and associated therapeutic effects. Mitochondrial transfer rates of (a) hMSC, (b) Pg-hMSC, (c) Fe-hMSC, and (d) Pg-Fe-hMSC at the indicated cell number ratios for co-culture (different engineered hMSCs:BLM-TC-1) (n = 3). Relative ROS levels of BLM-TC-1 after co-culture with (e) hMSC, (f) Pg-hMSC, (g) Fe-hMSC, and (h) Pg-Fe-hMSC at the indicated cell number ratios (n = 3). Cell viability of BLM-TC-1 after co-culture with (i) hMSC, (j) Pg-hMSC, (k) Fe-hMSC, and (l) Pg-Fe-hMSC at the indicated cell number ratios (n = 3). Data are presented as means \pm SD. Statistical significance was analysed using ordinary one-way ANOVA.

3. Figure 5 – authors refer to the organoid model as spheroid, however there is no lumen inside (according to the images), it looks like a cell aggregate. Not clear what physiological relevance this model has and how well it recapitulates features of the human IPF. It is not clear how MSCs were added to this aggregate – after induction of fibrosis or simultaneously, were they incorporated into the spheroid or migrated inside from the topical administration?

Response: Thank you for this valuable suggestion. Herein, the 3D fibrotic lung spheroid was prepared by coculturing human bronchial epithelial cells and lung fibroblasts on Matrigel. This multicellular spheroid model was reported to be used to reflect the human physiology of the drug-induced pulmonary fibrosis of human (*Biomed Mater.* 2022; 9:17(4)). We agree with the reviewer's correction that this multicellular spheroid model is more like a cell aggregate than an organoid model. We had corrected the description of this 3D fibrotic lung spheroid in the revised manuscript.

In addition, hMSCs were added to the spheroid model after the induction of fibrosis. We made several modifications in the Methods section to describe this experimental process more clearly. Furthermore, an additional experiment was supplied to show the migration of hMSCs into the fibrotic lung spheroid after topical administration (Supplementary Fig. S24).

Revision:

- **Abstract**

“Animal studies showed that such potent mitochondrial transfer using high-powered MSCs successfully mitigates fibrotic progression in a progressive PF model, which was further verified in **humanised multicellular lung spheroid models.**”

- **Supplementary Fig. S24**

Supplementary Fig. S24. Distribution of Pg-Fe-hMSC in the fibrotic 3D multicellular humanised lung spheroid. Fluorescent signals of Pg-Fe-hMSC were observed inside the 3D fibrotic lung spheroid (left panel). The distribution images were then reconstructed according to the fluorescent signals and were demonstrated in the right panel. Green: human bronchial epithelial cells and lung fibroblasts; red: Pg-Fe-hMSC.

- **Supplementary Methods**

“Determination of mitochondrial transfer rates and therapeutic effects on 3D fibrotic lung spheroid

Different engineered hMSCs were **added from the top to** the 3D fibrotic lung spheroid at the cell number ratio of 1:3 (hMSC:lung cell) **after induction of fibrosis.** The mitochondrial transfer rate and the therapeutic efficiency were quantified 24 h post the co-culture. The determination of mitochondrial transfer rates and therapeutic effects were similar as the determination of TC-1 cells.”

Reviewer #2

This paper by Huang and colleagues describes a mechanism of mitochondrial transfer from engineered human mesenchymal stem cells (hMSCs) to diseased cells as potential therapeutic strategy for pulmonary fibrosis (PF). The authors used a combination of in vitro, in vivo and ex vivo approaches to show that engineered hMSCs, treated with pioglitazone to increase the mitochondrial content, transfer mitochondria to recipient cells and have anti-fibrotic effect in the lung. Overall, the paper is not carefully written, and this limits the enthusiasm for the paper. The title is difficult to interpret and does not catch the attention of the readers.

Response: We would like to acknowledge and express our appreciation for the comments from the reviewer of our work. A careful revision of the manuscript had been performed, especially the title, the abstract, the introduction and the discussion had been reorganized according to the reviewer's valuable suggestions. We believe these modifications will significantly improve the manuscript. In addition, other concerns raised by the reviewer were addressed in the following.

1. The abstract is almost totally focused on the mitochondrial transfer mechanism. Although this is good, because this method is potentially applicable to other diseases with mitochondrial alterations, a more clear link with PF should be provided. Why PF? Why pioglitazone? Any links with PGC1 α dysfunction during PF? Why the comparison with pirfenidone? These questions should be addressed in the abstract and throughout the manuscript to connect the 2 topics of this paper. The introduction is lengthy, an idea could be to summarize background, significance and main results and move all the additional thoughts in the discussion section.

Response: A significant revision of the manuscript has been performed following your insightful suggestions, which includes the rewriting of the title, modification of the abstract, and simplification of the introduction. Considering that the critical roles of mitochondrial dysfunction in the progress of PF have been addressed in our previous study (*Sci Adv.* 2021;7(40):eabj0534), we herein focused more on promoting mitochondrial biogenesis to further augment

the mitochondrial transfer rate and the sustained transfer ability for efficient mitigation of progressive PF.

Revision:

- **Title**

“Efficient intervention for pulmonary fibrosis *via* a powerful mitochondrial transfer promoted by mitochondrial biogenesis”

- **Abstract**

“The use of exogenous mitochondria to replenish the damaged mitochondria in diseased cells has been proposed as a novel strategy for the intervention of pulmonary fibrosis (PF). However, the efficiency of this strategy is partially restricted by the current inability to supply sufficient mitochondria specifically to diseased lung cells. Herein, we report the artificial generation of high-powered mesenchymal stem cells (MSCs) co-engineered using pioglitazone and iron oxide nanoparticles (IONPs). The significantly promoted mitochondrial biogenesis by pioglitazone through the PGC-1 α pathway in addition to the Connexin 43 gap junctions facilitated mitochondrial transfer through IONPs stimulation, leading to a powerful mitochondrial transfer specifically targeting to the injured lung cells. Notably, for the first time highly efficient mitochondrial transfer is shown to not only restore mitochondrial homeostasis but also reactivates the inhibited mitophagy in injured cells, consequently recovering impaired cellular functions. Animal studies showed that such potent mitochondrial transfer using high-powered MSCs successfully mitigates fibrotic progression in a progressive PF model, which was further verified in a humanised multicellular lung spheroid model. The findings in the present study may provide a potential strategy to overcome the current limitations in mitochondrial replenishment therapy, thereby promoting therapeutic applications for PF intervention.”

- **Introduction**

“Mitochondria are the organelles that control the homeostasis, bioenergy, metabolism, biosynthesis, and apoptosis of eukaryotic cells¹⁻³. Recent studies have revealed additional functions of mitochondria as signalling organelles that participate in cell stress sensing, cell dynamics, protein secretion, and cell ageing⁴⁻⁸. Such critical roles of mitochondria in functioning cells make

mitochondrial dysfunction becoming a considerable pathogenesis⁹⁻¹¹. For example, several previous studies have indicated that the dysfunction of mitochondria in lung cells is one of the main causes of promoting pulmonary fibrosis (PF)^{12,13}. In addition, a recent study further has shown that mitochondrial quality has a significant impact on PF progression¹⁴. Therefore, the repair of dysfunctional mitochondria has been proposed as a novel strategy for disease intervention¹⁵. Among all the strategies available to correct the mitochondrial functions, mitochondrial replenishment therapy (MRT), which uses exogenous mitochondria to replenish dysfunctional mitochondria, is one of the most promising. Accumulating evidence has supported the effectiveness of MRT in reconstructing the damaged mitochondrial homeostasis and restoring cellular functions^{16,17}.

Highly efficient mitochondrial delivery with high specificity towards mitochondrial dysfunctional cells is the key to the success of MRT. To date, mitochondrial delivery approaches can be broadly categorised into direct and indirect delivery routes¹⁵. The former one involves directly transplanting isolated mitochondria into the disease sites locally, which poses the challenges such as the lack in selectivity in identifying mitochondrial dysfunctional cells, and the fast deactivation of the isolated mitochondria in the extracellular environment¹⁵. Conversely, an indirect delivery route exploits a more recently discovered ability to spontaneously donate mitochondria from normal cells to injured cells^{17,18}, showing the major advantages of avoiding the additional isolation of mitochondria and protecting the activity of transferred mitochondria. Thus, this naturally occurring phenomenon provides a new avenue for using living cells as both a donor and carrier of mitochondria for MRT.

However, a major challenge in the indirect therapeutic strategy is finding optimal mitochondrial carrier cells, which should have the ability to migrate to diseased tissues and recognise dysfunctional cells, and initiate intercellular mitochondrial delivery. Furthermore, it is preferable that these carrier cells have low bioenergetic needs and possess the powerful ability to generate mitochondria, which supports efficient and continuous mitochondrial exportation. Nevertheless, such a demand seems to counteract the nature of cells. Generally, cells with powerful mitochondrial generation capacities, such as

myocardial cells, also have high bioenergetic requirements¹⁹. Conversely, mesenchymal stem cells (MSCs) have low bioenergetic needs in their glycolytic state^{20,21}, making them more willing to donate their mitochondria to bioenergy desiderated cells^{17,22}. However, the low energy demands of MSC often restrict their mitochondrial generation, thereby limiting the number of mitochondria available for transportation and adversely affecting therapeutic outcomes. For example, we had previously observed an efficient intervention for PF using MRT with MSCs engineered by iron oxide nanoparticles (IONPs). However, these engineered MSCs showed poor therapeutic efficiency against the progressive PF, partly due to the insufficient and quickly exhausted mitochondrial transfer²³. In fact, the importance of an efficient and continued supply of mitochondria to restore the mitochondrial bioenergetics of injured lung epithelial cells (LECs) has been described in the interventions for PF²⁴.

The present study aimed to determine the possibility of promoting the mitochondrial biogenesis in low energy demanding MSCs, as well as the therapeutic benefits of these mitochondrial increased MSCs in the treatment of progressive PF. We observed for the first time that careful treatment of human placental-derived MSC (hMSC) with pioglitazone (Pg), a prescription drug for diabetes, could efficiently activate the mitochondrial biogenesis in hMSC primarily through the PGC-1 α -NRF1-TFAM pathway. Furthermore, joint engineering using Pg treatment and IONPs stimulation (Pg-Fe-hMSC) could achieve highly efficient and sustained intercellular mitochondrial delivery of the engineered hMSC, which we termed high-powered hMSC. Consequently, these high-powered hMSCs demonstrated a prominent therapeutic ability to intervene in PF progression in a mouse model with progressive PF (Fig.1). Furthermore, the efficient therapeutic potential of the high-powered hMSCs was confirmed in both fibrotic human lung cells and three-dimensional (3D) humanised lung spheroids.”

2. In vivo methods to study PF are adequate, however it is not indicated why some readouts (such as collagen expression) have fewer biological replicates than the others. The entire paper is focused on epithelial cells. Although it is known that alveolar epithelial cells present mitochondrial alterations, other cell types in the fibrotic lungs, in particular pathogenic fibroblasts (see Caporarello

et al. Thorax 2019) manifest reduced expression of PGC1a and mitochondrial dysfunction, but this is not mentioned. What the authors think about mitochondrial uptake by other cell types? The engineered cells are injected intravenously, what about vascular endothelial cells? It would be interesting to further address which type is more prone to the mitochondrial uptake and the respective contributions to the anti-fibrotic effect.

Response: Thank you for this important suggestion. There were fewer biological replicates of collagen expression than in other evaluations because whole lung samples were required for homogenisation to determine the collagen expression levels. This means that once the lung samples were used to determine collagen expression, they could not be used for other evaluations, such as Masson trichrome staining. Similarly, once the lung samples were used for collecting bronchoalveolar lavage fluids, they could not be applied for other determinations. Therefore, we had to divide the lung samples into different groups for different examinations, which led to fewer biological replicates in some evaluations than in others. The replicates were set according to the required lung samples corresponding to the related determinations. We clarified this issue in the revised manuscript. In addition, to address the reviewer's concerns, we re-performed the therapeutic study to determine collagen expression with more replicates. Similar results as those shown in Fig. 4c were obtained in the inhibition of collagen expression after the indicated treatments, confirming the reliability of our results.

Fig. R1. Relative collagen expression in mice lung after the indicated treatments on Day 28 post the initial administration of BLM (n = 6). Data are presented as means ± SD. Statistical significance was analysed using ordinary one-way ANOVA.

Regarding mitochondrial uptake by different cell types, we think that all the mitochondrial dysfunctional cells theoretically tend to accept the donated mitochondria from MSCs. In our previous studies, we observed that the efficient intercellular mitochondrial transfer only happened when the epithelial cells were injured, showing a high selectivity for injured cells (Fig. R2) (*Sci Adv.* 2021;7(40):eabj0534). Therefore, we propose that only injured lung cells accept a mitochondrial transfer. To confirm this hypothesis, we further evaluated the mitochondrial transfer rate from hMSCs to healthy or injured cells, including lung epithelial cells (TC-1), lung fibroblasts (Hs888Lu), and human umbilical vein endothelial cells (HUVECs). The results showed a significant intercellular mitochondrial transfer rate in all injured cells, meanwhile, all healthy cells almost did not receive exogenous mitochondria (Supplementary Fig. S9). Interestingly, regarding cell types, it was observed that epithelial cells demonstrated the highest mitochondrial transfer rate, fibroblasts showed a slightly lower mitochondrial transfer rate, and endothelial cells had the lowest mitochondrial transfer rate. However, all injured cells showed a much higher mitochondrial transfer rate than healthy cells. Therefore, the mitochondrial

transfer rate was more determined by cell injury than by cell type. This characteristic largely prevents off-target delivery of mitochondria to healthy cells such as the healthy vascular endothelial cells.

Fig. R2. Injured cells-selected mitochondrial transfer using MSCs (n = 3). ** $P < 0.01$. (*Sci Adv.* 2021;7(40):eabj0534).

Herein, we focused on pulmonary epithelial cells because the epithelial injury is widely recognised as the initiating event of fibrosis (*Lancet.* 2012;380(9842):680-8. *Annu Rev Pathol.* 2022;17:515-46. *J Clin Invest.* 2020;130(10):5088-99), and epithelial cells are key cells in the pulmonary environment (*Lancet.* 2011;378(9807):1949-61. *Nature.* 2021;595(7865):114-9). However, as suggested by the reviewer, other lung cell types such as lung fibroblasts are also involved in fibrosis. Therefore, additional evaluations of the mitochondrial transfer from different engineered hMSCs toward the bleomycin-treated fibroblasts (Hs888Lu) are supplied in the revised manuscript (Supplementary Fig. S11). Similar to the results with epithelial cells, the injured fibroblasts also received exogenous mitochondria and Pg-Fe-hMSCs demonstrated the highest transfer efficiency. Moreover, such mitochondrial transfer was also observed in injured human umbilical vein endothelial cells (HUVECs). These results show that only injured cells accept exogenous mitochondria delivered by MSCs. Several modifications based on the aforementioned results have been made in the revised manuscript to address the issues highlighted by the reviewer.

Revision:

● Results, Line 306

“Hematoxylin and eosin (H&E) staining and Masson’s trichrome staining of lung sections (harvested from three mice sample) further confirmed the optimal

therapeutic potential of Pg-Fe-hMSC, which remarkably reduced the fibrotic areas (Fig. 4b, left panel). Furthermore, the quantitative analysis according to the Masson's trichrome staining further verified the significantly relieved fibrotic ratio by Pg-Fe-hMSC (Fig. 4b, right panel). Moreover, three lung samples in each treatment group were homogenized to determine the collagen expression levels, showing a marked decrease in collagen deposition by Pg-Fe-hMSC (Fig. 4c). The determination of bronchoalveolar lavage fluid (BALF) collected from ten lung samples further showed significantly reduced expression levels of MMP9 (Fig. 4d) and TGF- β (Fig. 4e) after the treatment with Pg-Fe-hMSC. Similarly, measurement of hydroxyproline from eight homogenized lung samples confirmed the best therapeutic efficiency of Pg-Fe-hMSC (Fig. 4f), which was more efficient than Fe-hMSC or Pg-hMSC."

● **Supplementary Fig. S9**

Supplementary Fig. S9. Intercellular mitochondrial transfer from hMSCs to different receiving cells. (a) Mitochondrial transfer rate from hMSCs to healthy and injured TC-1 cells (n = 3). (b) Mitochondrial transfer rate from hMSCs to healthy and injured Hs888Lu cells (n = 3). (c) Mitochondrial transfer rate from hMSCs to healthy and injured human umbilical vein endothelial cells (n = 3). Data are presented as means \pm SD. Statistical significance was analysed using two-sided Student's t-test.

● Supplementary Fig. S11

Supplementary Fig. S11. Mitochondrial transfer rates of different engineered hMSCs.

(a) Mitochondrial transfer rates of different engineered hMSCs to BLM-treated Hs888Lu cells ($n = 3$). (b) Mitochondrial transfer rates of different engineered hMSCs to BLM-treated human umbilical vein endothelial cells (HUVECs) ($n = 3$). Data are presented as means \pm SD. Statistical significance was analysed using ordinary one-way ANOVA. ns, no significant difference.

3. An important paper in the field by Yu and colleagues (Nature Medicine 2018)

was cited to support the in vivo model of PF, but it was not discussed. In this paper, the Authors demonstrated that the thyroid hormone has an anti-fibrotic effect by promoting mitochondrial biogenesis, improving mitochondrial bioenergetics and attenuating apoptosis in alveolar epithelial cells both in vitro and in vivo (in a bleomycin model of PF) through PGC1a. What are the thoughts of the Authors in this regard? How the injection of engineered hMSCs and the transfer of mitochondria may have a better translational value compared to using a drug that mimic the same effects?

Response: The cited reference provides important support for our strategy, where the normalisation of mitochondrial function is crucial for the intervention of pulmonary fibrosis. Recently, an increasing number of studies highlighted the critical roles of mitochondria in manipulating cell behaviour and determining cell fate. Therefore, diverse strategies have been developed to maintain mitochondrial homeostasis, including the use of mitochondrial protective drugs and the supply of exogenous mitochondria. Thus far, it is difficult to say which strategy has a better translational value, because this field is still at an early stage and both strategies have several challenges yet to be surmounted. In our view, the mitochondrial transfer strategy using hMSCs has the advantage of inherently selecting mitochondrial dysfunctional cells, which means that only the mitochondrial homeostasis in injured cells will be affected. This issue is difficult to address with drug treatment without the assistance of an injured cell-targeted drug delivery system. Therefore, we think that our strategy has the potential for an efficient and precise restoring of mitochondrial homeostasis. However, the efficiency of mitochondrial transfer between cells is the major limitation in this strategy, especially compared with the direct improvement of mitochondrial bioenergetics using drugs. Therefore, we are focusing on the improvement of the intercellular mitochondrial transfer rate. Several discussions regarding the cited references and our thinking about this issue have been added to the revised manuscript. Thank you again for your valuable suggestions.

Revision:

● **Introduction, Line 90**

“However, these engineered MSCs showed poor therapeutic efficiency against the progressive PF, partly due to the insufficient and quickly exhausted

mitochondrial transfer²³. In fact, the importance of an efficient and continued supply of mitochondria to restore the mitochondrial bioenergetics of injured lung epithelial cells (LECs) has been described in the interventions for PF²⁴.”

- **Discussion, Line 537**

“Furthermore, compared to the previous strategy, which applied thyroid hormone to directly promote the mitochondrial biogenesis in LECs to improve mitochondrial bioenergetics and attenuate mitochondrial-regulated apoptosis, thus inhibiting PF²⁴, our current MRT strategy possesses an attractive superiority due to the inherent selection of mitochondrial dysfunctional cells. Therefore, we believe that the described strategy involving Pg-Fe-hMSC has the potential for an efficient and precise regulation of mitochondrial homeostasis.”

References

24. Yu, G., et al. Thyroid hormone inhibits lung fibrosis in mice by improving epithelial mitochondrial function. *Nat Med* **24**, 39-49 (2018).

Reviewer #3:

Summary:

The article by Huang et al. is a follow-up to their previous study published in 2021. In the previous study, they evaluated the role of iron oxide nanoparticles to augment gap junctions for improving mitochondrial transfer. In this study, they tested the hypothesis of increasing mitochondrial biogenesis to improve mitochondria replacement therapy (MRT); transferring mitochondria from a donor cell such as hMSC (human mesenchymal stem cell) to an injured lung cell. In this study, they first used a bleomycin-injured mouse TC-1 alveolar epithelial cell line to show proof of MRT *in vitro*. Pioglitazone (Pg) was used to induce biogenesis successfully, and when combined with iron oxide particles, they observed a synergistic effect with significantly improved mitochondrial transfer (Pg-Fe-hMSCs). They go on to show that Pg biogenesis works through the PGC-1 α /NRF1/TFAM pathway, using siRNA and plasmid, for knockdown and upregulation experiments respectively. Having successfully shown mechanistic effects in an *in vitro* model using mouse alveolar epithelial cells (AEC), they move on to testing their joint-engineered therapy (Pe-Fe-hMSCs) in a rodent mouse model of pulmonary fibrosis. Once again, they see similar results in the rodent model, where the joint-engineered therapy performed better compared to individual-engineered therapies in preventing fibrotic lung disease progression. Significantly, in this study, lungs treated with joint-engineered therapy show (i) better morphology (ii) lower collagen accumulation, and (iii) reduced fibrotic markers (i.e., TGF). Then they show similar results in fibrosis models developed using human airway cells and spheroids. Finally, they show that their treatment can reverse mitophagy inhibition induced by bleomycin. Together, the authors present a novel alternative treatment modality for pulmonary fibrotic lung disease.

Response: We greatly appreciate the reviewer's kind recognition of our work, as well as the valuable comments and careful correction. Several modifications including the throughout revision of the typing errors, figure improvements and the quantitative verifications of the 3D model of PF have been made in the revised manuscript in response to those helpful suggestions. We think the comments and suggestions proposed by the reviewer significantly improved our manuscript.

Revisions:

Significant revisions need to be made to this article for it to be published. My comments are below.

1. L64 – “.....spontaneous mitochondrial donation from the normal to dysfunctional cells.”. Dysfunctional cells could be injured or genetically abnormal. Please explain your choice of words here. Perhaps it is better to say injured here, as the entire article

Response: Thank you for this valuable suggestion. Herein, the dysfunctional cells indicate the mitochondrial dysfunctional cells, which can be caused by physical or chemical or biological damage. Therefore, we used “dysfunctional cells” to describe these cells with injured mitochondria. In the present study, the dysfunction of mitochondria was caused by the bleomycin-induced injury, thereby, “injured cells” are applied to describe these cells treated with bleomycin. To avoid this confusion, we corrected the description of these cells as “injured cells” throughout the manuscript.

Revision:

● **Introduction, Line 70**

“Conversely, an indirect delivery route exploits a more recently discovered ability to spontaneously donate mitochondria from normal cells to injured cells^{17,18}, showing the major advantages of avoiding the additional isolation of mitochondria and protecting the activity of transferred mitochondria.”

2. L83 – “...transfer mitochondria selectively to the dysfunctional cells that require more energy.” Liu et al. and Gomzikova et al., both state that healthy cells can transfer mitochondria to each other during physiologic conditions. Do you think Pg-hMSCs could increase mitochondrial transfer to healthy cells? Is there a threshold beyond which cells will transfer their mitochondria? Have you done any experiments looking at the transfer of mitochondria between healthy cells? Liu, D., Gao, Y., Liu, J. et al. Intercellular mitochondrial transfer as a means of tissue revitalization. Sig Transduct Target Ther 6, 65 (2021). <https://doi.org/10.1038/s41392-020-00440-z> Gomzikova MO, James V,

Rizvanov AA. Mitochondria Donation by Mesenchymal Stem Cells: Current Understanding and Mitochondria Transplantation Strategies. *Front Cell Dev Biol.* 2021 Apr 7;9:653322.

Response: Thank you for this interesting question. We also previously noticed the conditions of the intercellular mitochondrial transfer and made some explorations. As demonstrated in our previous study, we observed a significantly augmented intercellular mitochondrial transfer rate when we co-cultured hMSCs with injured TC-1 cells, whereas such transfer occurred at a very low rate between healthy hMSCs and healthy TC-1 cells (Fig. R1) (*Sci Adv.* 2021;7(40):eabj0534). This result indicates that the mitochondrial transfer may occur between healthy cells but its efficiency is very low.

Fig. R1. Injured cells-selected mitochondrial transfer using MSCs (n = 3). ** $P < 0.01$. (*Sci Adv.* 2021;7(40):eabj0534).

We think that this injured cell-selected mitochondrial transfer provides a significant advantage for using hMSCs as the mitochondrial donor cells, largely avoiding the off-target delivery of mitochondria to healthy cells. We included additional experiments in the revised manuscript to support this injured cell-selected mitochondrial transfer using hMSCs. As shown in the revised Supplementary Fig. S9, the mitochondrial transfer from hMSCs was observed in all injured cells (lung epithelial cells, lung fibroblasts, and human umbilical vein endothelial cells) irrespective of cell type. In addition, the mitochondrial transfer from hMSCs to all kinds of healthy cells was negligible. Therefore, we believe that mitochondria transfer mostly occurs from healthy cells to injured cells. This transfer can also be detected between healthy cells, but the transfer rate is negligible.

Considering the extremely low mitochondrial transfer rate between hMSCs

and healthy cells, we believe that Pg-hMSCs would hardly increase mitochondrial transfer to healthy cells. To confirm this, we inserted additional data in the revised manuscript showing a low mitochondrial transfer rate of both hMSCs and Pg-hMSCs to healthy TC-1 cells (Supplementary Fig. S10).

Regarding the threshold of mitochondrial transfer, it is difficult to provide a specific number based on our current technology. We previously evaluated the upper limit of the mitochondrial transfer rate of hMSCs. It was observed that the greater the injury of receptor cells, the more efficient the mitochondrial transfer. However, the transfer rate has an upper limit of approximately 10 % for hMSC and 30 % for Fe-hMSC (Fig. R2) (*Sci Adv.* 2021;7(40):eabj0534). This is why we investigated in the present study whether increasing mitochondrial biogenesis would further enhance the mitochondrial transfer rate.

Fig. R2. Mitochondrial transfer rate from hMSC and Fe-hMSC to TC-1 cells treated with different concentrations of BLM ($n = 3$). Data are presented as means \pm SD. Statistical significance was calculated *via* ordinary two-way ANOVA. ** $P < 0.01$. (*Sci Adv.* 2021;7(40):eabj0534).

Additional data and the related discussions in response to the reviewer's concerns were inserted in the revised manuscript.

Revision:

- Supplementary Fig. S9

Supplementary Fig. S9. Intercellular mitochondrial transfer from hMSCs to different receiving cells. (a) Mitochondrial transfer rate from hMSCs to healthy and injured TC-1 cells (n = 3). (b) Mitochondrial transfer rate from hMSCs to healthy and injured Hs888Lu cells (n = 3). (c) Mitochondrial transfer rate from hMSCs to healthy and injured human umbilical vein endothelial cells (n = 3). Data are presented as means \pm SD. Statistical significance was analysed using two-sided Student's t-test.

● **Supplementary Fig. S10**

Supplementary Fig. S10. Limited mitochondrial transfer rate between healthy cells. Mitochondrial transfer rate from hMSC or Pg-hMSC towards healthy TC-1 cell (n = 3). Data are presented as means \pm SD. Statistical significance was analysed using two-sided

Student's t-test. ns, no significant difference.

3. L 98 – “demobnstrated” should be demonstrated. Typo

Response: Thank you for the correction. A double-check of the manuscript had been performed to sure no typo errors.

Revision:

Introduction, Line 98

“However, these engineered MSCs **demonstrated** poor therapeutic efficiency against the progressive PF, partly due to the insufficient and quickly exhausted mitochondrial transfer²³”

4. L 110 – “we thereby naming it as high powered MSC”. Incorrect grammar. Should be – we thereby named it as high powered MSC – or – we are thereby naming it as high powered MSC.

Response: The sentence had been corrected according to your kind suggestion. In addition, a throughout check of the typing and grammar errors had been performed by a native speaker.

Revision:

● **Introduction, Line 105**

“which we **termed** high-powered hMSC.”

5. L114 – Especially for a disease like Idiopathic Pulmonary Fibrosis (IPF) where there are only 2 FDA-approved drugs, and when both drugs are available for research through vendors (e.g., selleckchem), why didn't the authors try Nintedanib? What is the rationale for choosing Pirfenidone over Nintedanib for this study?

Response: In the present study, we first observed the satisfactory therapeutic capacity of Pg-Fe-hMSCs to prevent PF progress. To confirm this therapeutic potential, we further compared it with the FDA-approved drug Pirfenidone in our

mouse model of PF. Because both the therapeutic mechanism and administration route of Pirfenidone were different from those of our strategy, this comparison only aimed to exclude the interference caused by the animal models. Because Nintedanib has similar therapeutic mechanism and effects as Pirfenidone (*Respir Res.* 2021;22(1):268; *Am J Respir Crit Care Med.* 2019;200(2):168-174), we herein only compared the therapeutic effects with one of the two FDA-approved drugs. The comparison followed a previous report that used Pirfenidone only as the positive control (*Sci Adv.* 2020;6(22):eaba3167). Another reason for our choice of Pirfenidone was the relatively low costs. Nintedanib is associated with higher costs for medications and medical visits, as well as a higher global costs than Pirfenidone (*Respir Med Res.* 2022;83:100951). Additionally, in clinical studies, Pirfenidone has a relatively lower discontinuation rate than Nintedanib. For all the reasons above, we used Pirfenidone for the comparison. However, the reviewer provided a good suggestion, and we will compare the therapeutic efficiency with Nintedanib in the future. Several discussions about the selection of pirfenidone were added to the revised manuscript.

Revision:

- **Discussion, Line 531**

“Of note, the therapeutic efficacy of Pg-Fe-hMSC with two-dose administrations was even comparable with Pirfenidone treatment with multiple-dose administrations (21 consecutive days), and negligible treatment-induced side effects were observed. In the present study, we did not compare therapeutic efficacy with the other FDA-approved drug Nintedanib for PF treatment, because these two drugs share similar efficiency^{48,49}.”

References

48. Dempsey, T.M., Sangaralingham, L.R., Yao, X., Sanghavi, D., Shah, N.D. & Limper, A.H. Clinical Effectiveness of Antifibrotic Medications for Idiopathic Pulmonary Fibrosis. *Am J Respir Crit Care Med* **200**, 168-174 (2019).
49. Marijic, P., Schwarzkopf, L., Schwettmann, L., Ruhnke, T., Trudzinski, F. & Kreuter, M. Pirfenidone vs. nintedanib in patients with idiopathic pulmonary fibrosis: a retrospective cohort study. *Respir Res* **22**, 268 (2021).

6. L116 – Use either artificial or engineered to talk about the modified MSCs. For example, “....the present study provides a promising strategy of using engineered hMSC to obtain mitochondrial...”. Unsure if using both adds more information. If you feel otherwise, please justify usage.

Response: We agree with the reviewer’s suggestion. Description of the modified MSCs is unified as “engineered hMSC” in the revised manuscript.

7. L117 - Hence, the present study provides a promising strategy of using artificial (“artificial..”; typo) engineered hMSC to obtain (“the” should be deleted) mitochondrial carrier cells with high mitochondrial biogenesis and efficient intercellular mitochondrial delivery capabilities, which can be used as a powerful mitochondrial donor cells and vehicles for treating mitochondrial dysfunctional diseases. Typos are mentioned in parentheses. The underlined statement is superfluous. You already mention that these cells are mitochondrial carrier cells, and they have efficient delivery capabilities. No need to mention again that they are powerful donor cells and vehicles.

Response: Thank you for this kind suggestion. Considering the superfluous statement of this sentence, we removed this description in the revised manuscript.

8. L123 – joint-enigeerd should be joint-engineered. Typo

Response: This typing error had been corrected in the revised manuscript. Thanks for your careful correction.

Revision:

● **Fig. 1, Line 111**

“Fig. 1. Schematic illustration showing the strategy of using a **joint-engineered** mesenchymal stem cell (Pg-Fe-hMSC) to enable a powerful and sustained intercellular mitochondrial delivery specialised to injured lung epithelial cells (LECs) for the potent intervention of pulmonary fibrosis (PF).”

9. L125 – specilaised should be “specialized”

Response: Thank you for the correction. We used British English throughout the manuscript and kept the spelling as “specilaised”.

10. L131 – “...with autonomous navigation ability to target mitochondrial dysfunctional AEC after systemic administration...”. My question is whether the AECs are specifically targeted? It doesn’t appear to be so. This sentence is misleading.

Response: As we responded in Question 1, our previous study and supplied data in the revised manuscript supported the injured cells-selected mitochondrial transfer with hMSC. We agree with the reviewer’s comments that this sentence cannot describe this capacity precisely. Therefore, this sentence was modified in the revised manuscript.

Revision:

● **Figure 1, Line 118**

“The Pg-Fe-hMSC can be used as both a mitochondrial generation factory and a smart vehicle for selective mitochondrial transfer to injured lung cells, showing the advantages of enabling efficient and sustained mitochondrial transfer.”

11. L146 - Supplementary Figure 1 – You need to mention whether MFI is the median or mean fluorescence index. In the figure caption, you do say that it is mean +/- SD. So, it does appear to be the mean. However, the expansion of the acronym MFI should be mentioned somewhere within the manuscript or the supplementary document. After a quick search, I did not find this.

Response: Thank you for pointing out this important issue. Herein, MFI refers to mean fluorescence index. Expansion of the acronym MFI had been supplied in the revised manuscript. This revision can also be found in the response for next question.

12. L146 – Regarding Supplementary Figure 1 – It would help to have a label within the graph artwork that says (a) 1-day treatment, and (b) 5-day treatment.

Response: Thank you for this helpful suggestion. The time labels had been added in the revised manuscript.

Revision:

- **Supplementary Fig. 1**

Supplementary Fig. S1. Optimisation of using the treatment with Pg to increase the mitochondrial mass in hMSC. (a) Mitochondrial mass indicated by the mean fluorescence index (MFI) of the mitochondrial tracker in hMSCs after the treatment with Pg at different concentrations for 1 day ($n = 3$). (b) Mitochondrial mass in hMSCs after the treatment with Pg at different concentrations for five consecutive days ($n = 3$). Data are presented as means \pm SD. Statistical significance was analysed using ordinary one-way ANOVA.

13. L195 – The Figure 2e panels could use color channel legends displayed within the figure, as there are multiple color channels. While this information is listed in the figure caption, it would be useful for the reader to be able to see this information within the image.

Response: Thank you for the suggestion. The information for each color channel had been supplied in the revised Fig. 2e.

Revision:

- **Fig. 2e**

Fig. 2. (e) Fluorescence images indicate the correlation between mitochondrial mass and mitochondrial transfer capacity of hMSC, Pg-hMSC, and siPGC-Pg-hMSC. Blue: nuclei, green: BLM-TC-1, red: mitochondria of hMSC, white arrows indicate transferred mitochondria. Scale bar, 50 μ m.

14. L206 – for panel figures 2l to 2o the authors should include the cell type for each of the graphs within the image. This would help to make side-by-side comparisons. This information is listed in the figure caption, but it should also be included within the image itself.

Response: We agree with the reviewer’s suggestion. Additional labels to indicate the cell type is supplied in the revised manuscript for a convenient reading.

Revision:

● **Figs. 2l-2o**

Fig. 2. Mitochondrial transfer rates of (l) hMSC, (m) Pg-hMSC, (n) Fe-hMSC, and (o) Pg-

Fe-hMSC at different time points (n = 3, for each). Data are presented as means \pm SD. Statistical significance was analysed using Student's t-test or ordinary one-way analysis of variance (ANOVA).

15. L184 – The videos and the still images of the videos in the form of supplementary figures 5 and 6 are great. While I can see red mitochondria in BLM-TC-1 cells, it is not very clear. Could the authors comment if there is a better way to show the transfer of mitochondria between two cells using live imaging?

Response: Thank you for this important suggestion. We arranged the images according to the timeline in order to show the dynamics of the mitochondrial transfer process. Because there are 20 images in Supplementary Figs. S5 and S6, the size of each image is restricted, leading to the difficulty of showing the mitochondrial transfer clearly in each image. To avoid this problem, we supplied additional enlarged images at a typical time point during mitochondrial transfer. Revised Supplementary Fig. S8 clearly shows the mitochondrial transfer from hMSC or Pg-Fe-hMSC to BLM-TC-1 cells (indicated by white arrows).

Revision:

- **Supplementary Fig. S8**

Supplementary Fig. S8. Enlarged images of the above dynamic transfer process of mitochondria 2.5 h after the co-incubation. Transferred

mitochondria from hMSC or Pg-Fe-hMSC to BLM-TC-1 cells were indicated by white arrows. Pg-Fe-hMSC had a higher mitochondrial transfer rate than hMSC. Green: BLM-TC-1; red: mitochondria of hMSC; white arrows indicate the transferred mitochondria. Scale bar, 10 μm .

16. L243-254 – This comment corresponds to image panels 3e to 3h and panels 3i to 3l. In the former series (3e to 3h), PGC-1 α is inhibited/downregulated, and based on the supplementary data, this is using siRNA. In the latter series (3i to 3l), PGC-1 α is upregulated using gene transfection. Correspondingly for the latter series of images, the graph's x-axis labels read "PGC-1 α -hMSC". This makes sense. For the former series of images, it is not clear why the authors decided to use the x-axis label "Pg-shMSC"? Does this signify treatment with si-RNA? The use of this specific x-axis label is confusing, and the reason behind its usage is unclear. It seems like the label should have been "siPGC-1 α -Pg-hMSC". Please explain. The same naming convention is also used for the image panel in Figure 2e.

Response: This is a good suggestion. We totally agree with the reviewer's opinion that "*siPGC-1 α -Pg-hMSC*" is more suitable than "*PGC-1 α -hMSC*" to indicate the PGC-1 α downregulated Pg-hMSC. However, taking the length of labels into consideration, we used "*siPGC-Pg-hMSC*" and "*PGC-hMSC*" to indicate the PGC-1 α downregulated and upregulated cells, respectively. All the related labels had been modified in the revised manuscript.

Revision:

- **Figs. 2e-2k**

Fig. 2. Increased mitochondrial biogenesis after pioglitazone treatment and enhanced intercellular mitochondrial transfer. (e) Fluorescence images indicate the correlation between mitochondrial mass and mitochondrial transfer capacity of hMSC, Pg-hMSC, and siPGC-Pg-hMSC. Blue: nuclei, green: BLM-TC-1, red: mitochondria of hMSC, white arrows indicate transferred mitochondria. Scale bar, 50 μ m. (f) Relative levels of mtDNA of hMSC, Pg-hMSC, and siPGC-Pg-hMSC (n = 3). (g) Quantitative analysis of mitochondrial transfer rates of hMSC, Pg-hMSC, and siPGC-Pg-hMSC (n = 3). (h) Relative PGC-1 α expression of hMSC before and after the gene transfection using PGC-1 α plasmid (n = 3). Relative (i) NRF1 and (j) TFAM expression of hMSC before and after the gene transfection using the PGC-1 α plasmid (n = 3, for each). (k) Quantitative comparison of mitochondrial transfer rates between hMSC and PGC-hMSC (n = 3). Data are presented as means \pm SD. Statistical significance was analysed using Student's t-test or ordinary one-way analysis of variance (ANOVA).

● **Figs. 3e-3l**

Fig. 3. *In vitro* therapeutic potential of Pg-Fe-hMSC in mouse LECs. Impacts of Pg treatment on (e) ATP levels, (f) intracellular ROS levels, (g) MMP levels, and (h) Cell viability of BLM-TC-1 cells ($n = 3$, for each). Impacts of PGC-1 α expression on (i) ATP levels, (j) intracellular ROS levels, (k) MMP levels, and (l) Cell viability of BLM-TC-1 cells ($n = 3$, for each). Data are presented as means \pm SD. Statistical significance was analysed using ordinary one-way ANOVA.

17. L264 – This comment is regarding supplementary figure 10 – please add a legend to figure panels (a to d; e to h; i to l). It would help to have the cell-type legends in the figure.

Response: We agree with this suggestion. Labels of the cell types had been added in the revised Supplementary Fig. S15.

Revision:

- **Supplementary Fig. S15**

Supplementary Fig. S15. Cell number ratios impact the mitochondrial transfer rates and associated therapeutic effects. Mitochondrial transfer rates of (a) hMSC, (b) Pg-hMSC, (c) Fe-hMSC, and (d) Pg-Fe-hMSC at the indicated cell number ratios for co-culture (different engineered hMSCs:BLM-TC-1) (n = 3). Relative ROS levels of BLM-TC-1 after co-culture with (e) hMSC, (f) Pg-hMSC, (g) Fe-hMSC, and (h) Pg-Fe-hMSC at the indicated cell number ratios (n = 3). Cell viability of BLM-TC-1 after co-culture with (i) hMSC, (j) Pg-hMSC, (k) Fe-hMSC, and (l) Pg-Fe-hMSC at the indicated cell number ratios (n = 3). Data are presented as means \pm SD. Statistical significance was analysed using ordinary one-way ANOVA.

18. L297 – “treating” should be treated. Typo

Response: Thank you for this correction. The grammar errors had been

corrected.

Revision:

● **Results, Line 295**

“Another reason is that TC-1 cells **treated** with BLM at high concentrations showed a tendency to receive more exogenous mitochondria than those treated with BLM at low concentrations (Supplementary Fig. S16).”

19. L309 – This comment pertains to Figure 4a. Why does the CT of the lung for the hMSC group look translucent as opposed to other groups? Please explain.

Response: Thank you for this important comment. The mentioned issue is largely caused by the progressive changes in lung architecture owing to the relatively poor therapeutic efficiency of non-engineered hMSCs. Similar Micro-CT images of fibrotic lungs were described in several previous studies (e.g., *Heliyon*. 2023;9(3):e13598. *Respir Res*. 2010;11(1):181. *Phytother Res*. 202;35(10):5883-98). For example, the translucent lung images caused by high dosage treatment with bleomycin (BLM (H)) are shown in Fig. R3. Translucent lung images were also observed in fibrotic lungs, which resulted from the progressive changes in lung architecture (Fig. R4). In addition, an expert was invited to evaluate the CT images and confirmed our conclusions.

Fig. R3. Translucent Micro-CT image of fibrotic lungs after treatment with a high dosage of bleomycin (BLM (H), red box). (*Heliyon*. 2023;9(3):e13598).

Fig. R4. Micro-CT image of fibrotic lungs that became translucent as opposed to the control group (red box). (*Respir Res.* 2010;11(1):181).

20. L313 – why are statistical analyses and comparisons in Figure panels 4c-f made with the NT group? However, in Figure panels, 3a-d statistical analyses and comparisons are made with the hMSC group and not the NT group. Please explain.

Response: Thank you for this critical question. Herein, the NT group was used as negative control group to evaluate the therapeutic potential of different engineered hMSCs. However, all the treatment groups in Figs. 3a-3d showed a good therapeutic effect in comparison with the NT group. Therefore, we further compared the statistical differences between different engineered hMSCs to show the therapeutic advantages of Pg-Fe-hMSC. Based on the consistency of analyses between the figures, we revised the comparison with the NT group in Figs. 3a-3d with the detailed *P* value, which can also reflect the differences between engineered and non-engineered hMSC.

Revision:

- **Figs. 3 a-3d**

Fig. 3. *In vitro* therapeutic potential of Pg-Fe-hMSC in mouse LECs. (a) ATP levels, (b) intracellular ROS levels, (c) Mitochondrial membrane potential (MMP) levels, and (d) Cell viability of bleomycin-treated TC-1 cells (BLM-TC-1) after treatment with different engineered hMSCs (n = 3, for each). Data are presented as means ± SD. Statistical significance was analysed using ordinary one-way ANOVA.

21. L324 – The transmission electron microscopy image comparisons do not accurately show what is conveyed by the authors. That is, it does not show that Pg-Fe-hMSCs successfully reduced the proportion of abnormal mitochondria. Adding supplementary figure 13 is appreciated, but it still doesn't seem to show the described pattern.

Response: Thank you for this valuable comment. The assessment of mitochondrial damage using transmission electron microscopy (TEM) was performed as previously described (*Science. 2015;350(6265):aad0116*). This morphological evaluation has been considered as an important way to assess mitochondrial injury (*Nat Rev Mol Cell Biol. 2020;21(4):204-24. Mol Cell. 2023;83(6):857-876*). In addition, the TEM images were not only used for the morphologic evaluation, but also for quantitative measurements of the area and perimeter of mitochondria (*Science. 2015;350(6265):aad0116*). The demonstrated results, calculated from a large number of samples (n = 59-97), accurately show abnormal mitochondria (revised Fig. 4 h, i), which corresponds to the results of the morphological evaluation (revised Fig. 4j). Therefore, we believe that combining morphological evaluation with the determination of mitochondrial area and perimeter allows the proportion of healthy mitochondria to be correctly estimated. To show the results more clearly, we labelled the healthy and abnormal mitochondria with different colours (green and brown indicate healthy and impaired mitochondria, respectively) based on the morphologic evaluation (e.g., swollen cristae structures and disrupted

membranes). In addition, results of the mitochondrial area and perimeter are moved to the front of the morphological evaluation for demonstrating the quantitative results before the observations.

Revision:

● **Figs. 4g-4j**

Fig. 4. Therapeutic potential of Pg-Fe-hMSC in the mouse PF model. (g) Observations of mitochondrial morphology using TEM. Scale bars, 2 µm. Green indicates healthy mitochondria and brown indicates impaired mitochondria. (h) Area of mitochondria (n = 59-97) measured according to TEM observations. (i) Perimeter of mitochondria calculated according to TEM observations (n = 59-97). (j) Ratios of healthy mitochondria according to morphological observation (n = 6). (k) Representative lung images following H&E staining and Masson's trichrome staining after the indicated treatments. Scale bar, 200 µm. Data are presented as means ± SD. Statistical significance was analysed using ordinary one-way ANOVA.

22. L377 – The lung images of Pg-Fe-hMSC in figures 4k and 4b look very different. The former seems to have almost normal airspace as opposed to the latter. Also, based on Masson's staining there seems to be high variability between these two experiments. Please comment on the level of variability observed when treated mice using Pg-Fe-hMSCs.

Response: Thank you for this important comment. We think that the mentioned variance is mainly caused by the individual differences of mice in two batches

of experiments. Therefore, we previously set both Healthy group and NT group as the control groups to avoid interference from individual differences. The therapeutic efficiency of Pg-Fe-hMSCs was confirmed in both Figs. 4b and 4k in comparison with the NT group. In addition, we quantitatively evaluated the fibrotic ratio in the Healthy, NT, and Pg-Fe-hMSCs groups in Figs. 4b and 4k, and no statistically significant difference was observed (Fig. R5). Moreover, we asked a pathologist to re-evaluate our section samples, who confirmed the observed variations. He agrees that the differences in Masson's staining may be caused by different colour contrast in two batches of experiments. Therefore, we believe that the mentioned variability is caused by a batch difference and is acceptable.

Fig. R5. Comparison of the fibrotic ratio of the indicated group based on Masson's trichrome staining shown in Figs. 4b and 4k. The fibrotic ratio was calculated by measuring the percentage of the blue area within the pulmonary sections (n = 18, six fields of view from three different sections in each group).

23. L365 – “assistes” should be assists. Typo

Response: This error had been corrected in the revised manuscript and the similar errors were also be double-checked.

Revision:

● **Results, Line 370**

“Mitochondrial biogenesis **contributed** to the efficient treatment of PF in the humanised fibrotic models”

24. L368 – The authors claim that they have harvested human (alveolar epithelial cells) AECs from abandoned lung tissues using EpCAM antibodies. Unfortunately, this is incorrect and misleading. Alveolar epithelial cells refer to pure populations of cells of either type I or type II or type I&II cells. Based on the technique used for isolation, the authors have isolated small airway epithelial cells (SAEC), and these can include bronchiolar epithelial cells, basal stem cells, and alveolar epithelial cells (type I and II). It is incorrect to call these **alveolar epithelial cells**, especially when these cells have not been characterized. Please modify the manuscript to reflect these changes. Alternatively, I am happy to receive justification from your end regarding the naming scheme of **AECs**. In Supplementary Figure 16, the same cells are called human lung epithelial cells. Please also make sure to have consistent nomenclature in both the manuscript and supplementary materials.

Response: We appreciate for this professional correction. We had revised the description of these cells as lung epithelial cells (LECs) in both the revised manuscript and supplementary information.

Revision:

All the names had been revised from alveolar epithelial cells (AECs) to **lung epithelial cells (LECs)**.

25. L388 – the use of cell lines to create a PF model is commendable. However, in this case, the results presented by the authors are semi-quantitative at best. Immunofluorescence staining shows α -SMA, vimentin, collagen type I, and ki-67. Quantitative results for these markers are missing. Without this information, the authors cannot make a conclusive statement that they have induced PF phenotype in these cells. Please provide additional data.

Response: Thank you for this professional suggestion. Additional data of the semi-quantitative evaluation of these markers had been supplied in the revised manuscript.

Revision:

- **Supplementary Fig. 23**

Supplementary Fig. S23. Confirmation of the fibrotic 3D multicellular humanised lung model. The expression levels of α-SMA, vimentin, collagen-I, and Ki-67 were calculated using western blotting analysis. (n = 3). Data are presented as means ± SD. Statistical significance was analysed using ordinary one-way ANOVA.

26. L399 – “monocellualr” should be monocellular.

Response: This typing error had been corrected in the revised manuscript. Thank you again for the correction.

Revision:

- **Results, Line 406**

“Overall, Pg-Fe-hMSC demonstrated significantly high intercellular mitochondrial transfer capability, which could potentially be used against

fibrosis in both **monocellular** and multicellular humanised fibrotic models.”

27. L481 – In this first line of the discussion, it is unclear if the authors are talking about intracellular (within cell transfer) or intercellular (between cells) transfer of mitochondria. I believe this should have been intercellular. Please explain your choice of term.

Response: Thank you for this critical correction. Herein, we want to say the intercellular transfer of mitochondria.

Revision:

- **Discussion, Line 488**

“The findings of the critical roles of natural **intercellular** mitochondrial transfer in self-tissue repair^{18,40,41} have inspired a novel therapeutic strategy for restoring the energy metabolism of injured cells by introducing exogenous mitochondria obtained from other cells”

REVIEWERS' COMMENTS

Reviewer #1 (Remarks to the Author):

All my comments have been addressed, thank you!

Reviewer #2 (Remarks to the Author):

The manuscript is significantly improved compared to its original version.

The Authors addressed all my concerns.

Reviewer #3 (Remarks to the Author):

The authors have answered all my questions and they have also made satisfactory changes to the manuscript. I have no further questions or suggestions, and recommend the article be published.

Point-by-point Response to the Reviewers' Comments

Remarks to the Author:

Reviewer #1:

All my comments have been addressed, thank you!

Response: We greatly appreciate the reviewer's valuable comments. We believe that these modifications significantly improve our manuscript.

Reviewer #2

The manuscript is significantly improved compared to its original version.

The Authors addressed all my concerns.

Response: We would like to acknowledge and express our appreciation for the valuable comments from the reviewer of our work. We believe that these modifications significantly improve the manuscript.

Reviewer #3:

The authors have answered all my questions and they have also made satisfactory changes to the manuscript. I have no further questions or suggestions, and recommend the article be published.

Response: We greatly appreciate the reviewer's kind recognition of our work, as well as the kind advice. We think that the comments and suggestions proposed by the reviewer significantly improve our manuscript.